# Basement membrane damage by ROS- and JNK-mediated Mmp2 activation drives macrophage recruitment to overgrown tissue

Neha Diwanji [1] & Andreas Bergmann [1 ✉]

Macrophages are a major immune cell type infiltrating tumors and promoting tumor growth and metastasis. To elucidate the mechanism of macrophage recruitment, we utilize an overgrowth tumor model ("undead" model) in larval *Drosophila* imaginal discs that are attached by numerous macrophages. Here we report that changes to the microenvironment of the overgrown tissue are important for recruiting macrophages. First, we describe a correlation between generation of reactive oxygen species (ROS) and damage of the basement membrane (BM) in all neoplastic, but not hyperplastic, models examined. ROS and the stress kinase JNK mediate the accumulation of matrix metalloproteinase 2 (Mmp2), damaging the BM, which recruits macrophages to the tissue. We propose a model where macrophage recruitment to and activation at overgrowing tissue is a multi-step process requiring ROS- and JNK-mediated Mmp2 upregulation and BM damage. These findings have implications for understanding the role of the tumor microenvironment for macrophage activation.

[1] University of Massachusetts Medical School, Department of Molecular, Cell and Cancer Biology, 364 Plantation Street, LRB 419, Worcester, MA 01605, USA. ✉email: andreas.bergmann@umassmed.edu

Tumor-associated macrophages (TAMs) are one of the major immune cells that infiltrate the tumor and provide an inflammatory environment important for cancer progression[1]. They play a role in tumor growth, invasion, angiogenesis, and immunosuppression, thus helping in tumor survival and metastasis[1]. Often, apoptotic tumor cells release signals that attract circulating monocytes to tumors where they undergo alternative activation to yield M2-polarized TAMs[2–5]. The tumor microenvironment also helps in TAM recruitment[4,6], although not much is known about which cells of the microenvironment contribute to this process. While the identities of several key molecular and cellular players in TAM recruitment are well-established, there still remains a need to better understand the dynamic interactions that occur between the tumor and its microenvironment during this complex immune response.

*Drosophila melanogaster* has become a valuable model system to dissect the fundamental, mechanistic, and conserved aspects of inflammatory responses[7]. Several studies have reported the presence of macrophages in different *Drosophila* tumor models[8–13]. These macrophages, termed hemocytes, have tumor promoting as well as tumor inhibiting functions, analogous to mammalian macrophages. One such overgrowth-promoting role of hemocytes was recently described for apoptosis-induced proliferation (AiP)[14].

AiP is a form of compensatory proliferation where caspase-initiated signaling in apoptotic cells drives proliferation of neighboring healthy cells, thus maintaining tissue homeostasis following significant cell loss[15,16]. Early work in *Drosophila* has revealed that the Caspase-9-like initiator caspase Dronc is critical for at least one type of AiP[15,16]. To decipher the signaling pathways important for AiP, caspases are activated by expression of the IAP-antagonist *hid*, but the final step of cell death execution is blocked by co-expression of the effector caspase inhibitor *p35*, thereby keeping the cells in a permanent apoptotic state that is referred to as "undead". Because P35 specifically inhibits the activity of effector caspases, Dronc can persistently signal for AiP causing overgrowth of the undead tissue[17,18]. If *hid* and *p35* are co-expressed using the *ey-Gal4* driver (*ey > hid,p35*), the undead *ey-Gal4*-expressing portion of the larval eye imaginal discs (Supplementary Fig. 1c) and the undead adult heads are overgrown[17].

Using the undead model, we reported recently that active Dronc stimulates the transmembrane NADPH oxidase Duox for generation of extracellular reactive oxygen species (ROS) which attract hemocytes leading to activation of the stress kinase JNK (c-Jun N-terminal Kinase)[14]. JNK triggers the release of the mitogens Wg, Dpp, and EGF to induce AiP[17,19,20] as well as induces *hid*, setting up an amplification loop that continuously signals in undead cells for AiP[14,21].

The *Drosophila* larval eye-antennal imaginal disc gives rise to the adult eyes and head. It is made up of two juxtaposed epithelial cell layers, the columnar disc proper (DP) and the squamous peripodial epithelium (PE), which face each other apically separated by a lumen while their basal sides are exposed to the hemolymph and are surrounded by a basement membrane (BM) (Fig. 1a)[22]. Interestingly, hemocytes attached to eye discs are always observed at the basal side of the DP, not the PE. On control discs, hemocytes form large cell aggregates invariably posterior to the morphogenetic furrow (MF) (Supplementary Fig. 1a,b)[14]. There is also a hemocyte cell cluster on the antennal disc. However, on undead discs, hemocytes spread out to the undead tissue anterior to the MF, often as single cells, and extend cellular protrusions (Supplementary Fig. 1c)[14]. They release the TNFα-like ortholog Eiger[14] and additional hemocytes are recruited (Supplementary Fig. 1d). These differences in hemocyte behavior indicate that they are getting "activated" at undead discs. About half of the hemocytes on undead discs display the activated

morphology (Supplementary Fig. 1e). Similar observations were also made in the neoplastic *Ras^V12 scrib* tumor model[11]. However, the specific mechanisms of hemocyte recruitment to and activation at undead discs are unknown. It is possible that ROS produced by undead or tumor cells help to attract and activate circulating hemocytes to the overgrown tissue[11,14] as has been observed in the wound healing response in the *Drosophila* embryo[23]. In addition, damage to the BM following wounding or in *Ras^V12 scrib* tumors is sufficient for hemocyte recruitment in larvae[10,24].

The BM is a thin-layered polymer of matrix proteins that surrounds the basal side of epithelial tissues including imaginal discs (Fig. 1a)[25]. The BM is important for tissue shape, and provides a mechanical barrier preventing tumor cells from invading into distant tissues[26,27]. Major components of the BM are Collagen IVs, Laminins, Perlecan, and Nidogen[28,29]. These components are produced and secreted by fat body cells and hemocytes, and incorporated into the BM at imaginal discs and other tissues[28,30,31]. Matrix metalloproteinases (MMPs) are zinc-containing extracellular proteases that are critical for the reorganization of the BM[32]. The *Drosophila* genome encodes two MMP genes, *Mmp1* and *Mmp2*. Both MMPs have membrane-tethered and secreted forms[33–35].

Here, we report that the microenvironment of the overgrown undead tissue is important for recruiting and activating hemocytes. The BM of undead imaginal discs is severely damaged. This damage to the BM integrity correlates with generation of ROS and activation of JNK in undead tissue as well as in several neoplastic epithelial tumor models. However, ROS and JNK are not sufficient to cause BM damage in normal tissue. We find that Mmp2, but not Mmp1, is necessary and sufficient for disrupting the BM of undead as well as normal tissue. Both ROS and JNK regulate the expression and activity of *Mmp2*, and there is a strong accumulation of Mmp2 protein specifically at the basal side of DP cells. Finally, the damaged BM is sufficient for recruiting hemocytes from the circulation to the tissue. These data illustrate the influence of the tumor microenvironment for recruiting and activation of hemocytes to promote tumor growth.

## Results

**Undead discs have damaged BM.** Previous studies have implicated BM damage as a mechanism by which circulating hemocytes are recruited to wounds or tumors[10,23,24]. Therefore, we examined the integrity of the BM using antibodies that detect the BM components Perlecan and Laminin. In undead *ey > hid,p35* discs, the BM is severely disrupted compared to control (*ey-Gal4* and *ey > p35*) discs, which have an intact and uniform BM (Fig. 1b–d; Supplementary Fig. 2). There appear to be gaps, folds, and deposits of the BM around which hemocytes are attached (Fig. 1d). Damage at the surface of undead discs was also directly observed by scanning and transmission electron microscopy (SEM and TEM) (Fig. 1e–j). Importantly, BM damage was observed in all experimental discs, while all control discs displayed an intact BM (Supplementary Table 1). Interestingly, the irregularities of the BM occur only at the DP side of undead discs, while the BM on the PE side is not affected (Fig. 1d, g; Supplementary Figs. 2c; 3). Consistently, Perlecan and Laminin fluorescence are only reduced on the DP, but not PE, side of undead eye discs (Supplementary Fig. 3j,k). Based on these observations, we used the fluorescence on the PE side as an internal control for quantifying the changes in Perlecan and Laminin fluorescence on the DP side (Fig. 1l; Supplementary Fig. 2d). Because hemocytes are only present at the DP side of undead discs (Fig. 1a), these data imply that damaged BM might be responsible for recruiting hemocytes to these overgrown discs.

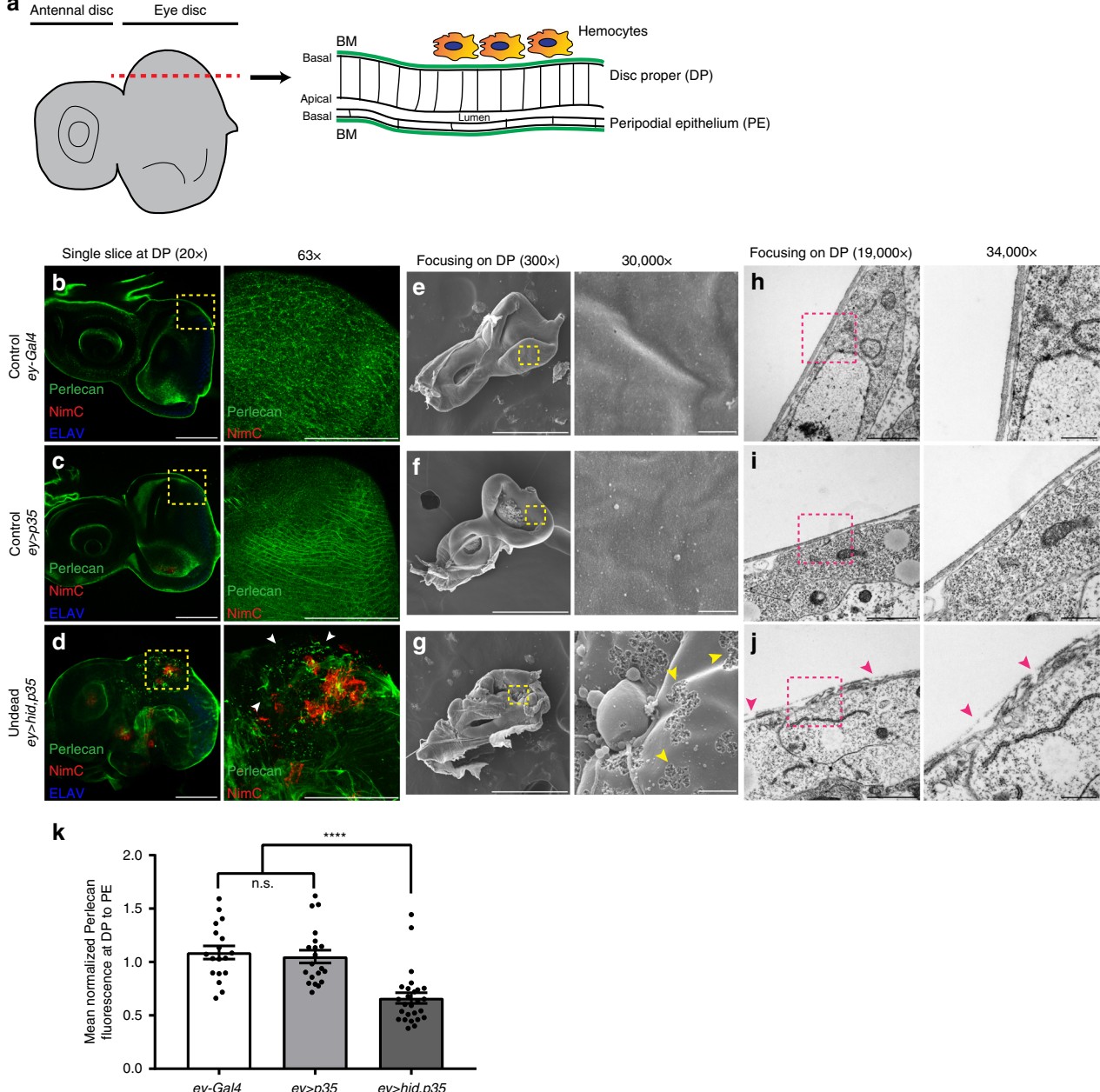

**Fig. 1 Undead discs have damaged basement membranes (BM). a** Schematic representation of a surface view (XY; left) and an orthogonal section view (XZ; right) along the red dotted line through a 3rd instar larval eye-antennal imaginal disc. The columnar disc proper (DP) and squamous peripodial epithelium (PE) are apically separated by a lumen, their basal surfaces surrounded by the basement membrane (BM, green). Hemocytes attach specifically at the basal side of the DP. **b–d** Representative examples of control (ey-Gal4 (**b**) and ey > p35 (**c**)) and experimental (undead) (ey > hid,p35 (**d**)) eye discs labeled for the BM marker Perlecan (green), for hemocytes with NimrodC (NimC) (red) and ELAV (blue) marking posterior photoreceptor neurons. Single slices focusing on the basal side of the DP (left), yellow squares are magnified (right). Damaged BM surrounded by hemocytes (red) indicated by white arrowheads (**d**, right). Scale bars, 100 μm. **e–g** Scanning electron microscopy (SEM) images of ey-Gal4 (**e**), ey > p35 (**f**), and ey > hid,p35 (**g**) discs focusing on the DP side, yellow squares magnified (right). Damaged surface of BM indicated by yellow arrowheads (**g**, right). Scale bars, 200 μm (left) and 1 μm (right). **h–j** Transmission electron microscopy (TEM) images of ey-Gal4 (**h**), ey > p35 (**i**), and ey > hid,p35 (**j**) discs focusing on the DP side, magenta squares magnified (right). Discontinuous BM indicated by magenta arrowheads (**j**, right). Scale bars, 0.5 μm (left) and 0.2 μm (right). **k** Quantification of Perlecan labeling at DP normalized to PE as internal control reveals that the BM is damaged in undead ey > hid,p35 discs compared with controls. Quantification of Perlecan fluorescence is taken from single slices at the basal surface of DP and PE. Data represented as mean fluorescence ± SEM analyzed by one-way ANOVA with Holm–Sidak test for multiple comparisons. ****$p < 0.0001$, n.s. = no statistical significance. Data from $n = 18$ (ey-Gal4), 20 (ey > p35), and 26 (ey > hid,p35) discs analyzed from five independent experiments. Source data are provided as a Source Data file.

Next, we examined whether BM damage is triggered simply by activation of apoptosis. Because expression of *hid* without *p35* using *ey-Gal4* causes early larval lethality, we expressed *hid* using *spalt-Gal4* (*sal$^{EPv}$ > hid*) in wing imaginal discs. Although the BM

in these experiments does not appear to be as smooth as in controls, it is intact without any gaps, and the Perlecan intensity is comparable to control discs (Supplementary Fig. 4a–b,d). However, under undead conditions (*sal$^{EPv}$ > hid,p35*), the BM

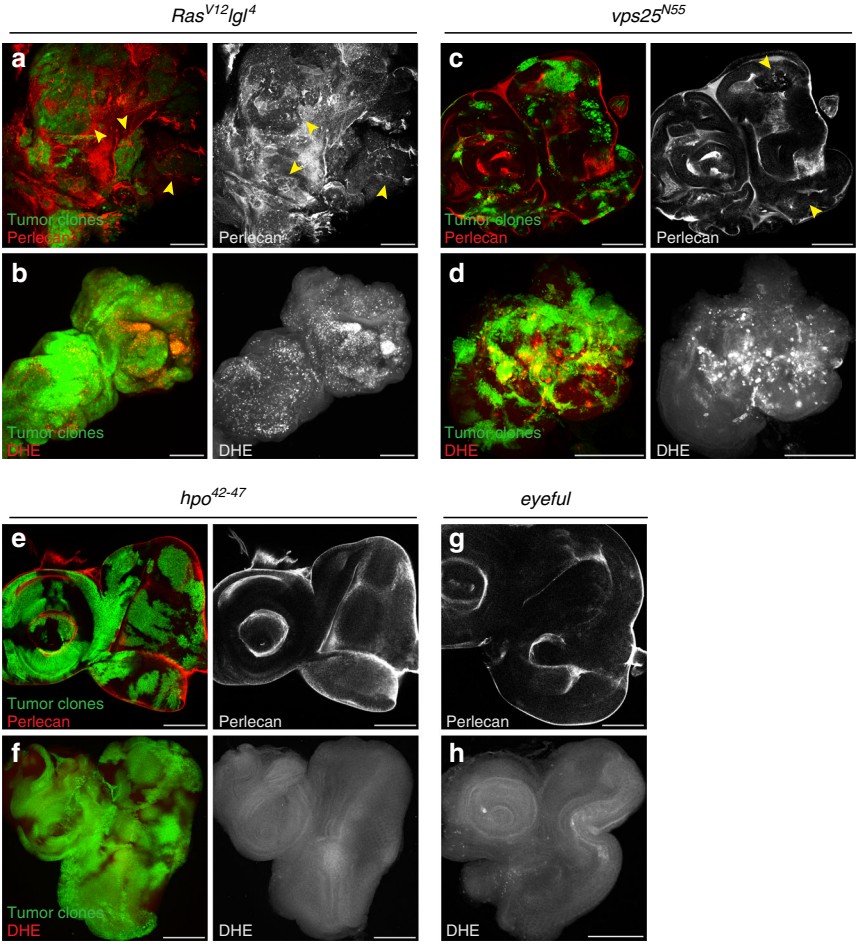

**Fig. 2 BM damage coincides with ROS production in neoplastic tumor models.** Except for *eyeful* (**g**, **h**), tumor clones were generated by the MARCM technique using *ey-FLP* and are marked by GFP. *eyeful* is uniformly expressed in the *ey-Gal4* domain. Scale bars, 100 μm. **a**, **c**, **e**, **g** The BM is detected by anti-Perlecan labeling (red left; gray right and **g**). Panels are single slices focusing on the basal side of DP. Quantification in Supplementary Fig. 6a. **b**, **d**, **f**, **h** ROS is detected by the dihydroethidium (DHE) indicator dye (red left; gray right and **h**). Panels are maximum intensity projections. Quantifications in Supplementary Fig. 6b. Exact genotypes: **a**, **b** *yw ey-FLP/+; lgl⁴ FRT40A/tub-Gal80 FRT40A; act > y+ >Gal4, UAS-GFP56ST/+*. **c**, **d** *yw ey-FLP/+; FRT42D vps25^{N55}/FRT42D tub-Gal80; act > y+ >Gal4 UAS-GFP56ST/+*. **e**, **f** *yw ey-FLP/+; FRT42D hpo^{42-47}/FRT42D tub-Gal80; act > y+ >Gal4 UAS-GFP56ST/+*. **g**, **h** *yw ey-Gal4/+; UAS-Dl UAS-psq UAS-lola/+*.

now shows signs of damage with gaps and holes in the Perlecan labeling, and significant decrease in Perlecan fluorescence (Supplementary Fig. 4c,d). These data suggest that induction of apoptosis alone is not sufficient for BM damage. Instead, it is the undead condition that is responsible for the BM damage.

**BM damage coincides with ROS production in neoplastic tumors.** Because overgrown undead discs have a damaged BM, we wondered whether the BM damage is simply a consequence of the overgrowth. To address this question, we examined several neoplastic and hyperplastic overgrowth (tumor) models in *Drosophila*. First, the mosaic *Ras^{V12}lgl* condition is a neoplastic tumor model, characterized by unrestricted proliferation, failure to differentiate, tissue invasion, and death of the organism[36–39]. Mosaic *Ras^{V12}lgl* eye discs show BM disruption as visualized by gaps in Perlecan labeling (Fig. 2a; quantified in Supplementary Fig. 6a), similar to the related *Ras^{V12}scrib* condition[38,39]. Second, imaginal discs mosaic for the ESCRT-II component *vps25* are highly overgrown[40–42]. Examining the BM of this neoplastic tumor model, we observed disruptions of Perlecan labeling at mosaic discs indicating that there is BM damage in this tumor context (Fig. 2c; Supplementary Fig. 6a). Third, the Hippo pathway is an important regulator of cell proliferation and cell death, and

mutations in pathway components, like *hippo*, cause a hyperplastic phenotype characterized by increased proliferation, but leaving cellular architecture intact[37]. Interestingly, even though *hippo* mosaic discs are strongly overgrown, they do not show BM damage by Perlecan labeling. The overgrown discs have a uniform and intact BM and do not show any gaps in Perlecan labeling (Fig. 2e; Supplementary Fig. 6a). Finally, deregulation of the Notch pathway is associated with tumors in both humans and *Drosophila*, and *eyeful* represents a Notch-induced hyperplastic overgrowth model in *Drosophila* eye discs[43]. In overgrown *eyeful* discs, an intact BM is observed without any irregularities in Perlecan labeling (Fig. 2g; Supplementary Fig. 6a). Taken together, these data imply that BM damage is not a universal hallmark for all overgrowth models.

To identify commonalities in these tumor models that would result in differential effects on BM integrity, we measured levels of ROS. Elevated ROS levels are characteristic of multiple cancers and are thought to promote tumor growth[44]. Using the ROS indicator dihydoethidium (DHE), we found that neoplastic *Ras^{V12}lgl* mosaic discs have increased ROS levels (Fig. 2b; quantified in Supplementary Fig. 6b), similar to the related *Ras^{V12}scrib* model[11]. Likewise, *vps25* mosaic discs have elevated levels of ROS. Intense DHE puncta are observed in and around

tumor clones (Fig. 2d; Supplementary Fig. 6b). In contrast, hyperplastic *hippo* mosaics as well as *eyeful* discs do not produce any detectable ROS as indicated by the absence of DHE labeling in these overgrown discs (Fig. 2f, h; Supplementary Fig. 6b). Together, these data suggest that BM damage and ROS production coincide with the induction of neoplastic growth. Similarly, undead discs also demonstrate ROS production[14,45,46] and BM damage (Fig. 1), providing evidence for a link between ROS and BM damage in this overgrowth context as well.

In undead AiP, ROS directly or indirectly activate hemocytes followed by activation of JNK[14]. Therefore, we tested for activation of hemocytes and JNK in the overgrowth models. Consistently and as shown previously[36,40,47,48], $Ras^{V12}lgl$ and *vps25* mosaic discs are characterized by increased JNK signaling (Supplementary Figs. 5b,d, 6e). Furthermore, these neoplastic discs are associated with hemocytes of an activated morphology (Supplementary Figs. 5a,c, 6c,d). In contrast, although the hyperplastic *hippo* mosaics and *eyeful* discs are attached by a large number of hemocytes, these hemocytes form tight aggregates and only few are activated (Supplementary Figs. 5e,g, 6c,d). JNK is also not activated in these discs as indicated by the lack of pJNK labeling (Supplementary Figs. 5f,h, 6e).

Together, there is increased production of ROS, BM damage as well as activation of hemocytes and JNK in neoplastic tumor models like $Ras^{V12}lgl$ and *vps25*. In contrast, hyperplastic tumors caused by *hippo* mutants and *eyeful* lack these characteristics. Based on these observations, we propose that the undead $ey > hid,p35$ model represents a neoplastic model as it shows all the neoplastic characteristics that are absent in the hyperplastic models.

**ROS and JNK are required, but not sufficient for BM damage.** Our observations suggest that there is a correlation between ROS production and BM damage, and therefore we examined if ROS are required for BM damage. RNAi-induced knockdown of *Duox*, the NADPH oxidase which synthesizes ROS in undead discs[14], or overexpression of the secreted catalase *hCatS* results in loss of ROS and suppression of $ey > hid,p35$ overgrowth[14,45,46]. Under these ROS-depleted conditions, BM disruption was also suppressed (Fig. 3a–c, e; Supplementary Fig. 7a–c,e) suggesting that ROS are required for BM damage. Along with suppression of BM damage, loss of ROS also causes impaired hemocyte recruitment to the discs and loss of JNK activation[14].

Next, we determined if ROS are sufficient for BM damage. However, overexpression of *Duox* is not sufficient for ectopic ROS production[49]. Therefore, we exposed wild-type eye discs to hydrogen peroxide ($H_2O_2$) ex vivo, and monitored the effect on the BM. Exposure of discs to $H_2O_2$-containing media had no effect on the BM (Fig. 3f, g, l), suggesting that although ROS are required for BM damage, they are not sufficient to induce the damage.

Because activation of JNK is an important factor for AiP[14,17,19], we examined if JNK is necessary for BM damage. We blocked JNK activity by expressing a dominant negative construct ($JNK^{DN}$) in $ey > hid,p35$ discs and monitored the effect on BM integrity. Loss of JNK signaling prevents BM damage and Perlecan labeling is not disrupted (Fig. 3d, e). Similar results were obtained with Laminin labelings (Supplementary Fig. 7d,e) suggesting that JNK activation is required for BM damage.

To examine whether JNK activation was sufficient to damage BM, we expressed a constitutively active JNK kinase transgene ($hep^{CA}$) using $sal^{EPv}$-Gal4, as *ey-Gal4*-induced expression of $hep^{CA}$ causes early larval lethality. However, $sal^{EPv} > hep^{CA}$ wing discs, similar to the $sal^{EPv} > GFP$ control discs, are not characterized by BM damage (Supplementary Fig. 7f–h). Therefore, similar to ROS, JNK activation is necessary, but not sufficient for BM damage.

Finally, we tested whether $H_2O_2$ treatment of $sal^{EPv} > hep^{CA}$ discs can cause damage to the BM ex vivo. Similar to $H_2O_2$-treated eye discs (Fig. 3g), treatment of $sal^{EPv} > GFP$ control wing discs did not significantly damage the BM (Fig. 3i). However, while visual inspection of $H_2O_2$-treated $sal^{EPv} > hep^{CA}$ discs gives the impression of BM defects in the $sal^{EPv}$-expression domain (Fig. 3k), the quantification of the Perlecan labelings did not reveal any statistically significant difference (Fig. 3l). Combined, these data suggest that ex vivo, $H_2O_2$ and JNK together are not sufficient for BM damage suggesting that additional factors are involved in BM damage.

**Hemocytes do not damage the BM of undead discs.** To exclude the possibility that recruited hemocytes themselves cause BM damage, we analyzed the BM of undead discs in a heterozygous *serpent* (*srp*) background. *srp* encodes a GATA-type Zn finger transcription factor required for hemocyte differentiation[50]. $srp^{neo45}$ specifically affects hemocyte differentiation[51] and dominantly suppresses the overgrowth phenotype of $ey > hid,p35$ animals[14]. Despite this suppression, the BM of $ey > hid,p35$ discs heterozygous for $srp^{neo45}$ was disrupted with large gaps in Perlecan labeling (Supplementary Fig. 8a–c). Similar to control discs (Supplementary Fig. 1a,b), few hemocytes are present on these discs which form aggregates with very little cellular protrusions indicating a naïve morphology (Supplementary Fig. 8d–g). These results indicate that that there is no linear relationship between BM damage and the amount of hemocytes attached to the discs, and suggest that BM damage occurs largely independently of hemocytes.

**MMPs are upregulated in the undead discs.** Matrix metalloproteinases (MMPs) are well-known for their remodeling function of the extracellular matrix[32]. The *Drosophila* genome contains two MMP genes, *Mmp1* and *Mmp2*[34,52]. To determine if MMPs are involved in BM damage of undead discs, we first examined if they are expressed in undead discs. Previous studies have shown that JNK promotes upregulation of *Mmp1* in tumors as well as in the undead discs[14,17,19,20,53]. Consistently, we observe a marked upregulation of *Mmp1* transcripts in $ey > hid,p35$ discs compared with controls (Supplementary Fig. 9a). Likewise, *Mmp2* transcripts are also elevated in undead discs (Supplementary Fig. 9a), accompanied by increased Mmp2 protein levels (Fig. 4a–c). In control eye discs, Mmp2 protein is localized in the cells of the PE and apically in DP cells with little localization at the basal side (Fig. 4a, b, d). Strikingly, in undead discs, Mmp2 protein is not only upregulated at the PE (Supplementary Fig. 9b), but also shows a strong accumulation at the basal side of the DP (Fig. 4c, d), where the BM is preferentially damaged. Interestingly, Mmp2 protein levels are uniformly upregulated across the entire disc, even outside of the *ey-Gal4* expression domain (Fig. 4c; Supplementary Fig. 9c). *Mmp2* RNAi strongly reduces this uniform upregulation (Supplementary Fig. 9d,e) suggesting that uniform upregulation of Mmp2 is real and that the Mmp2 antibody is specific. Because Mmp2 is produced both as membrane-anchored and secreted proteins[35], the easiest explanation for the uniform distribution of Mmp2 in undead discs is the secreted nature of Mmp2. Interestingly also, there is a weak upregulation of Mmp2 in $ey > p35$ control discs (Fig. 4a, b; Supplementary Fig. 9a). However, despite this upregulation, there is no accumulation of Mmp2 protein at the basal side of the DP in $ey > p35$ discs (Fig. 4d).

**Mmp2 upregulation is dependent on ROS and JNK.** Because of the accumulation of Mmp2 at the basal side of the DP, and because—as shown below—only Mmp2, but not Mmp1, is

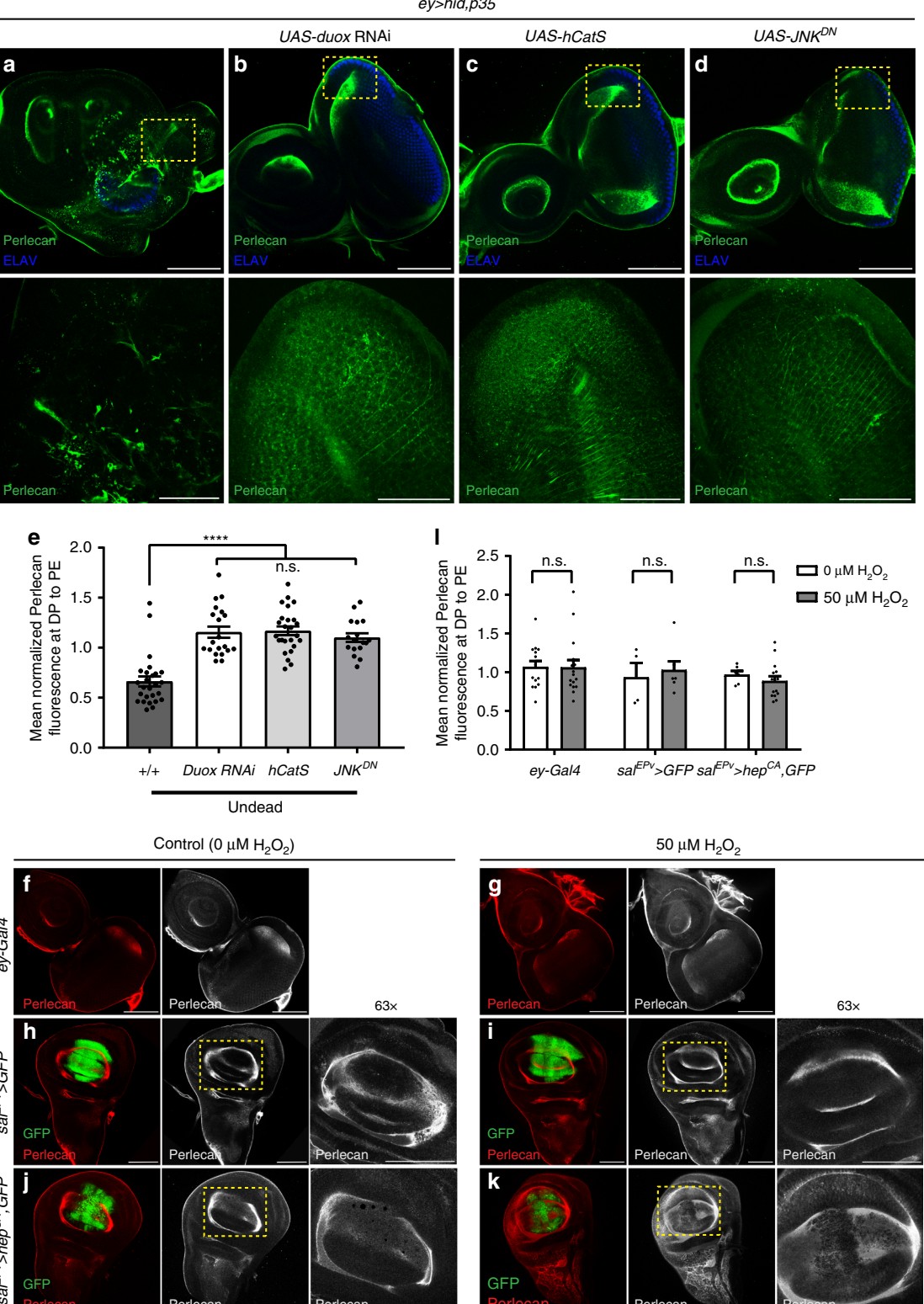

required for BM damage, we focused on Mmp2 in the following. Excitingly, upon loss of ROS or JNK in undead discs, the increased upregulation and accumulation of Mmp2 at the basal side of the DP is strongly reduced and is similar to control discs (Fig. 4e–g, h). These data imply that the upregulation of Mmp2 in undead discs is controlled by ROS and JNK, thereby placing Mmp2 downstream of them in the AiP pathway.

Next, we asked whether $H_2O_2$ treatment ex vivo and increased JNK activity, individually or together, are sufficient to upregulate Mmp2. $H_2O_2$ treatment of $sal^{EPv} > GFP$ control wing discs or hyper-activation of JNK ($sal^{EPv} > hep^{CA}$) did not affect Mmp2 levels (Supplementary Fig. 10a–c,e). Similarly, $H_2O_2$ treatment of $sal^{EPv} > hep^{CA}$ discs did not cause any detectable change of Mmp2 levels in the $sal^{EPv}$-expression domain

**Fig. 3 ROS and JNK are required, but not sufficient to cause BM damage. a–d** In *ey > hid,p35* eye imaginal discs, RNAi-induced depletion of ROS-producing *Duox* (**b**), transgenic overexpression of a secreted catalase *hCatS* (**c**) and inactivation of JNK by expression of dominant negative JNK (*JNK^DN*) (**d**) results in suppression of BM damage compared with undead controls (**a**). Single slices focusing on the basal side of DP (top); yellow squares are magnified below. Scale bars, 100 μm (top), 50 μm (bottom). **e** Quantification of Perlecan intensity in **a–d** is taken from single slices at the basal surface of DP and PE. Fluorescence intensity at PE was used as internal control for normalization. Data represented as mean fluorescence ± SEM analyzed by one-way ANOVA with Holm–Sidak test for multiple comparisons. ****$p < 0.0001$, n.s. = no statistical significance. Data from $n = 26$ (*ey > hid,p35*), 20 (*duox* RNAi), 25 (*hCatS*), and 17 (*JNK^DN*) discs analyzed from four independent experiments. **f, g** Representative examples of wild-type (*ey-Gal4*) eye imaginal discs incubated ex vivo in Schneider's media with 0 μM $H_2O_2$ (**f**) or 50 μM $H_2O_2$ (**g**) labeled for Perlecan (red or gray). Scale bars, 100 μm. **h–k** Representative examples of wing imaginal discs with wild-type (**h, i**) or hyper-activated JNKK (*hep^CA*) (**j, k**) incubated ex vivo in Schneider's medium with 0 μM $H_2O_2$ (**h, j**) or 50 μM $H_2O_2$ (**i, k**) and labeled for Perlecan. Yellow squares in middle panels are magnified at right. Scale bars, 100 μm. **l** Quantification of Perlecan intensity in **f–k** is taken from single slices at the basal surface of DP and PE. Fluorescence intensity at PE was used as internal control for normalization. Data represented as mean fluorescence ± SEM analyzed by one-way ANOVA with Holm–Sidak test for multiple comparisons. n.s. = no statistical significance. Data from $n = 14$ (*ey-Gal4* 0 μM $H_2O_2$), 16 (*ey-Gal4* 50 μM $H_2O_2$), 4 (*sal^EPv > GFP* 0 μM $H_2O_2$), 7 (*sal^EPv > GFP* 50 μM $H_2O_2$), 6 (*sal^EPv > hep^CA, GFP* 0 μM $H_2O_2$), and 15 (*sal^EPv > hep^CA,GFP* 50 μM $H_2O_2$) discs analyzed from three independent experiments. Source data are provided as a Source Data file.

(Supplementary Fig. 10d,e) suggesting that ex vivo, activation of JNK signaling and exposure to $H_2O_2$, either independently or together, are not sufficient to upregulate Mmp2. This result explains why $H_2O_2$ treatment of *sal^EPv > hep^CA* discs did not cause significant BM damage (Fig. 3k, l).

**Mmp2 has increased enzymatic activity in undead discs**. To examine whether the elevated Mmp2 levels in undead discs also correspond to increased enzymatic activity, we performed several enzymatic assays. Fluorescent dye-quenched-(DQ-)gelatin is a polymer-substrate that contains multiple quenched FITC conjugates. Cleavage of DQ-gelatin releases the green FITC fluorescence which can be analyzed in several ways[35,54]. First, in fluorometric assays of protein lysates from imaginal discs, we measured a significant fluorescence increase in *ey > hid,p35* discs compared with controls (Fig. 4i). This increased enzymatic activity in *ey > hid,p35* discs is dependent on *Mmp2* and similar to discs that overexpress *Mmp2* (*sal^EPv > Mmp2*) (Fig. 4j) suggesting that Mmp2 accounts for this increased cleavage activity toward DQ-gelatin. Furthermore, reduction of ROS and inhibition of JNK suppresses the enzymatic activity of Mmp2 in fluorogenic assays (Fig. 4k).

Second, we performed ex vivo in situ zymography assays with eye discs of different genotypes. In these assays, the DQ-gelatin substrate was incubated with dissected, unfixed discs and the fluorescence scored under the confocal microscope[54]. While control discs show a uniform, low-level fluorescence across the disc, we noted an increased enzymatic activity through release of fluorescence quenching at undead discs (Supplementary Fig. 11a–c). *Mmp2* RNAi restores the fluorescence quenching (Supplementary Fig. 11d) suggesting that Mmp2 accounts for this activity. Furthermore, the enzymatic activity of Mmp2 is dependent on JNK and ROS (Supplementary Fig. 11e–g). Interestingly, hemocytes attached to the discs also label strongly for this fluorescence marker (Supplementary Fig. 11b,c) suggesting that they also contain enzymatic activity toward DQ-gelatin.

Third, using gel zymography, where Mmp2 activity is recognized by loss of the in-gel gelatin substrate as visualized by clear band upon Coomassie staining[35,55], there is an increased gelatin cleavage activity in *ey > hid,p35* extracts which is suppressed by *Mmp2* knockdown and reduction of ROS or JNK activity (Supplementary Fig. 11h,i). Taken together, these data suggest that the increased Mmp2 protein levels at undead discs have an increased enzymatic activity in a JNK- and ROS-dependent manner.

**Mmp2 but not Mmp1 is required for BM damage and AiP**. After establishing that MMPs are upregulated and that Mmp2 has increased enzymatic activity in undead discs, we investigated whether MMPs play a role for BM damage. Knockdown of *Mmp1* by RNAi did not suppress the BM damage of undead discs

(Fig. 5a, b, d). *Mmp1* RNAi also did not affect hemocyte recruitment to the overgrown discs. High numbers of hemocytes are still observed on these discs and many have the activated morphology (Fig. 5e–f, h, i).

In contrast, *Mmp2* RNAi strongly suppressed the BM damage of *ey > hid,p35* discs. Perlecan labeling is intact and uniform around the discs (Fig. 5c, d). Similar data were also obtained with Laminin as BM marker (Supplementary Fig. 12). Knockdown of *Mmp2* also impaired hemocyte recruitment to the discs, and of the few hemocytes that are present, only very few showed the activated phenotype (Fig. 5g–i).

To gain insight into the functional requirement of MMPs for inducing overgrowth, we monitored the effect of *MMP* RNAi knockdown on the adult overgrowth phenotype of *ey > hid,p35* animals. While knockdown of *Mmp1* did not have any effect on overgrowth, *Mmp2* RNAi significantly suppressed the overgrowth of *ey > hid,p35* animals (Fig. 5j–m). Three independent RNAi transgenes targeting *Mmp2* gave consistent results (Supplementary Fig. 13a). Along with the effects on the adult overgrowth phenotype, hemocyte behavior and BM damage, knockdown of *Mmp2* also suppressed other AiP markers like ROS production and JNK activation (Supplementary Fig. 13b–g). Because ROS and JNK are required for *Mmp2* upregulation in undead discs (Fig. 4e–h), and in turn, Mmp2 controls ROS and JNK activity (Supplementary Fig. 13), suggests that Mmp2 participates in the feedback amplification loop which operates in undead cells to induce AiP and overgrowth[14]. Thus, Mmp2, but not Mmp1, is important for BM damage and hemocyte recruitment, and it participates in the feedback amplification loop to promote overgrowth of undead *ey > hid,p35* discs.

**Mmp2-mediated BM damage recruits hemocytes**. Next, we addressed if *Mmp2* is sufficient for damaging the BM. Because expression of *Mmp2* using *ey-Gal4* caused early larval lethality, we expressed it in wing discs using *sal^EPv-Gal4*. While over-expression of *Mmp1* had no effect on the BM (Fig. 6a–b; Supplementary Fig. 14a), expression of *Mmp2* severely damaged the BM. There was complete absence of Perlecan in the area where *Mmp2* was expressed (Fig. 6c, Supplementary Fig. 14a) consistent with previous studies[10]. The area of damage extended even beyond the Mmp2-expression domain, not just at DP, but even on the PE side (Fig. 6c; Supplementary Fig. 14a) consistent with the notion that Mmp2 has secreted isoforms. These results imply that Mmp2, but not Mmp1, is both necessary and sufficient to cause BM damage.

Finally, we examined if BM damage is sufficient to recruit hemocytes. In contrast to eye discs, there are no hemocytes attached to unstressed (control) wing discs (Fig. 6d). Likewise, hemocytes are not recruited to discs expressing *Mmp1* which

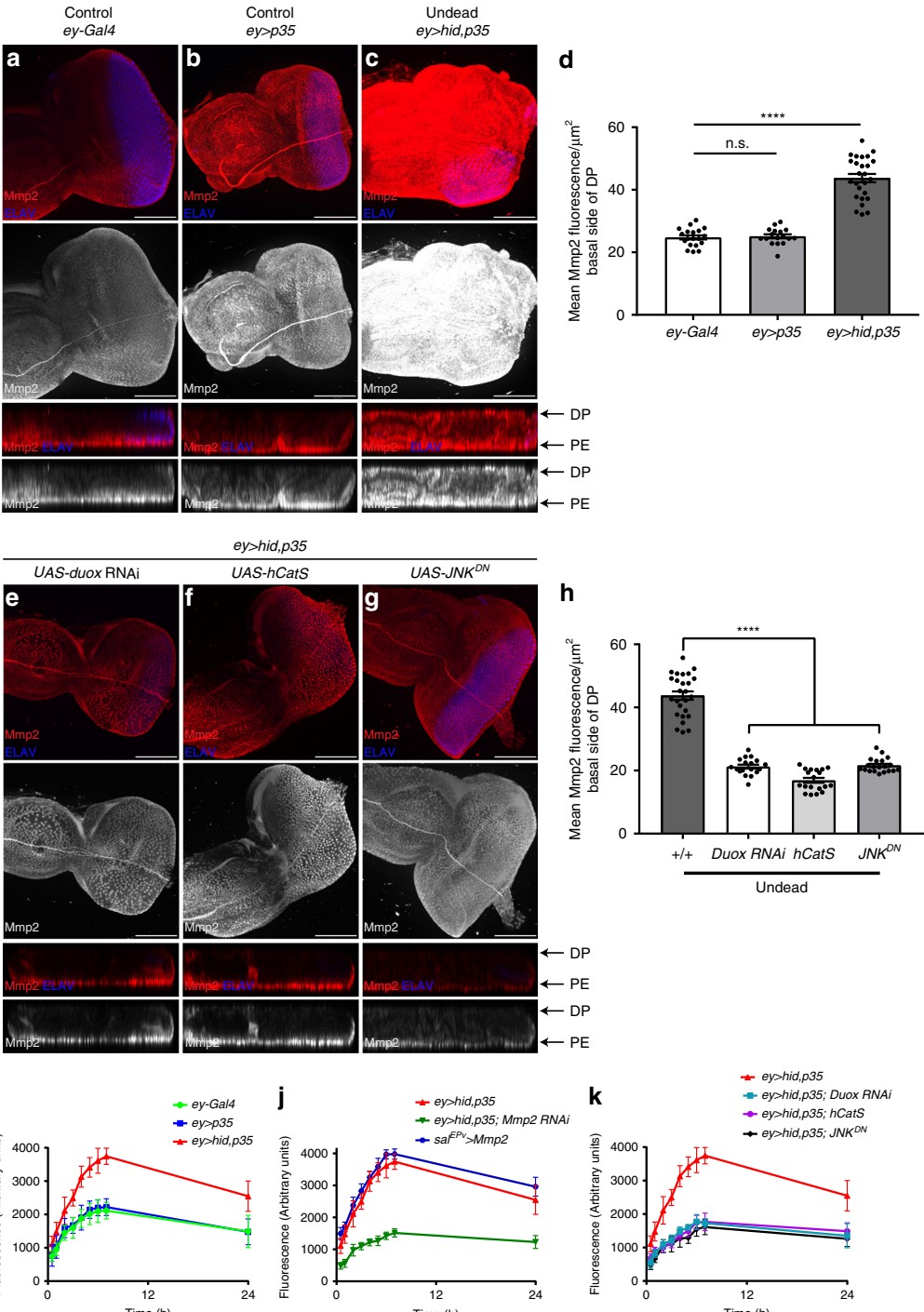

**Fig. 4 Upregulation of Mmp2 levels and activity in undead discs requires ROS and JNK. a–c** Representative examples of control (*ey-Gal4* (**a**) and *ey > p35* (**b**)) and undead (*ey > hid,p35* (**c**)) eye imaginal discs labeled for Mmp2 (red and gray) and ELAV (blue). Orthogonal (YZ) section views (bottom panels) show increased localization of Mmp2 at the basal side of the DP in undead *ey > hid,p35* discs compared with controls. Scale bars, 100 μm. **d** Quantification of Mmp2 intensity in **a–c** taken from single slices at the basal surface of DP, represented as mean fluorescence ± SEM analyzed by one-way ANOVA with Holm–Sidak test for multiple comparisons. ****$p < 0.0001$, n.s. = no statistical significance. Data from $n = 19$ (*ey-Gal4*), 17 (*ey > p35*), and 26 (*ey > hid,p35*) discs analyzed from four independent experiments. Quantification of Mmp2 intensity at the PE shown in Supplementary Fig. 9b. **e–g** Examples of *ey > hid, p35* eye imaginal discs expressing *Duox* RNAi (**e**), transgenic *hCatS* (**f**), and dominant negative JNK (*JNK^DN*) (**g**) labeled for Mmp2 (red and gray) and ELAV (blue). Orthogonal (YZ) cross section views (bottom) show suppression of the Mmp2 upregulation at the basal side of the DP compared with undead discs. Scale bars, 100 μm. **h** Quantification of Mmp2 intensity in **e–g** taken from single slices at the basal surface of DP, represented as mean fluorescence ± SEM analyzed by one-way ANOVA with Holm–Sidak test for multiple comparisons. ****$p < 0.0001$, n.s. = no statistical significance. Data from $n = 26$ (*ey > hid,p35*), 18 (*duox* RNAi), 19 (*hCatS*), and 20 (*JNK^DN*) discs analyzed in three independent experiments. **i–k** Mmp2 activity measured by fluorometric assay using DQ-gelatin cleavage substrate, represented as mean ± SEM from three independent experiments. The same *ey > hid,35* data (red lines) are shown in **i–k**. This increased fluorescence in *ey > hid,35* discs is lost upon knockdown of *Mmp2*, and is comparable to *Mmp2* overexpression (**j**). Mmp2 activity in *ey > hid,p35* undead discs is dependent on ROS and JNK (**k**). Source data are provided as a Source Data file.

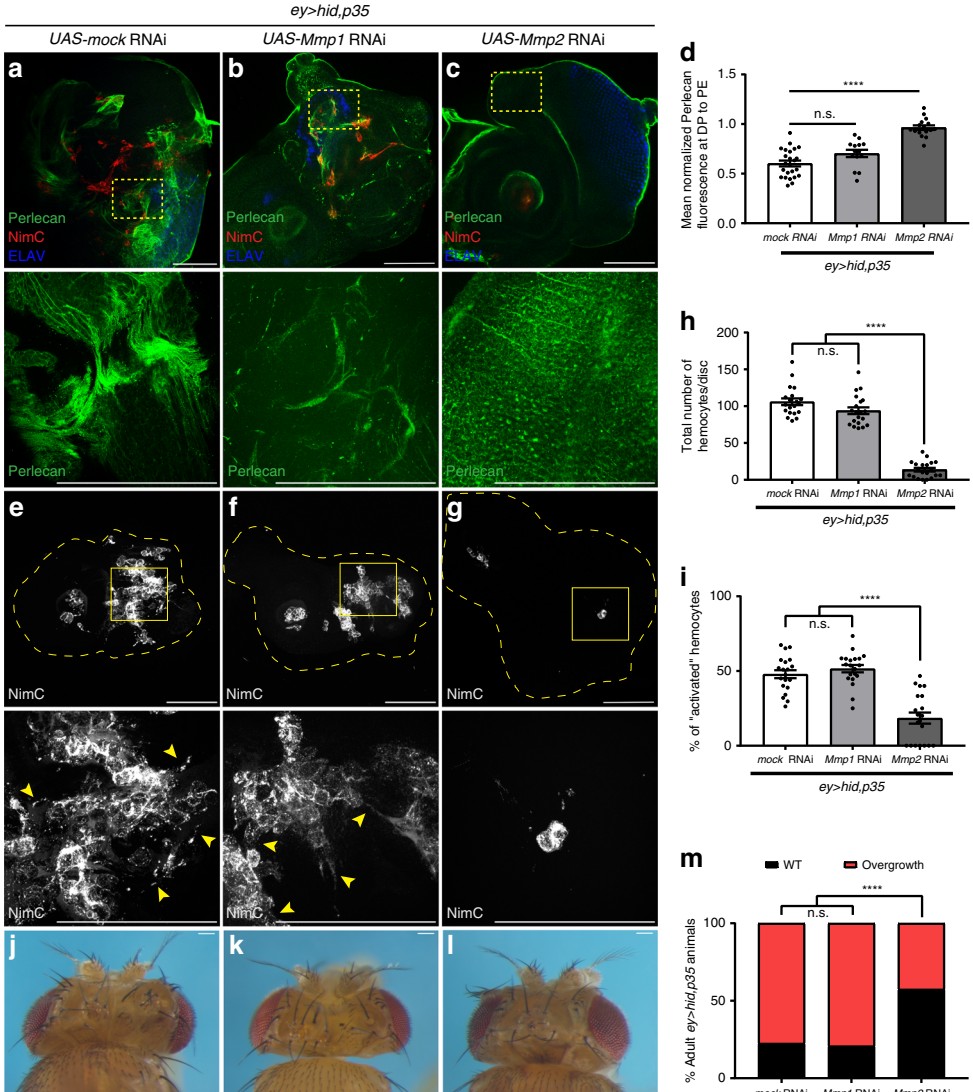

**Fig. 5 Mmp2, not Mmp1, is required for BM damage, hemocyte recruitment and AiP.** For all panels, scale bars 100 µm. ****$p < 0.0001$, n.s. = no statistical significance. **a–c** Single slices focused on the basal side of DP show uniform Perlecan labeling in *Mmp2* RNAi discs (**c**), while *Mmp1* RNAi discs (**b**) still show irregularities as in *mock* (*RFP*) RNAi discs (**a**). **d** Quantification of Perlecan intensity in (**a–c**) taken from single slices at the basal surface of DP and normalized to PE, represented as mean fluorescence ± SEM analyzed by one-way ANOVA with Holm–Sidak test for multiple comparisons. $n = 26$ (*mock* RNAi), 14 (*Mmp1* RNAi), 16 (*Mmp2* RNAi) discs analyzed from three independent experiments. **e–g** Recruitment and activation of hemocytes (NimC in gray) to *ey > hid,p35* discs is severely impaired upon knockdown of *Mmp2* (**g**), while *Mmp1* RNAi does not affect hemocyte recruitment (**f**) similar to *mock* RNAi discs (**e**). Yellow squares are magnified below. Yellow arrowheads (bottom) indicate activated morphology of hemocytes. **h, i** Quantification of hemocyte recruitment (**h**) and activation (**i**) in *ey > hid,p35* discs compared with *Mmp1* and *Mmp2* knockdown. Hemocyte activation was measured by the percentage of total hemocytes that had one or more cellular protrusions. Total number and percentage of activated hemocytes were analyzed by one-way ANOVA with Tukey's multiple comparisons test. $n = 20$ discs per genotype analyzed in three independent experiments. **j–l** *Mmp2* knockdown suppresses adult overgrowth. Representative images of adult heads of *ey > hid,p35* animals that are expressing either *mock* (*RFP*) RNAi (**j**), *Mmp1* RNAi (**k**), or *Mmp2* RNAi (**l**). *Mmp2* RNAi also suppresses larval imaginal disc overgrowth, as seen in **c**) and (**g**, where the disc morphology is restored, in contrast to *ey > hid, p35* discs in **a** and **e**. **m** Quantification of the suppression of adult *ey > hid,p35* overgrowth by *Mmp2* RNAi. Progeny was scored as wild type (WT) (black bars) or overgrown (red bars). Suppression is measured by a shift in percentage to WT from overgrown animals that is significantly different as determined by two-sided Fisher's exact test. $n = 100$ to 150 flies counted per genotype from four independent experiments. Source data are provided as a Source Data file.

have undamaged BM (Fig. 6e). However, high numbers of hemocytes are attached to discs with damaged BM due to expression of Mmp2 (Fig. 6f; Supplementary Fig. 14b) consistent with previous reports[10]. Interestingly, some of these hemocytes also display the activated morphology (Supplementary Fig. 14c,d). Importantly, expression of *Mmp2* did not lead to ectopic ROS production or caspase activation (Fig. 6g–l; Supplementary Fig. 14e,f), excluding the possibility that ROS or apoptosis are

responsible for the hemocyte recruitment to discs with damaged BM. Taken together, these findings demonstrate that BM damage alone is sufficient to recruit hemocytes to the discs.

## Discussion

In this study, we show a correlation between ROS generation and BM damage not only in the undead AiP model, but also in several

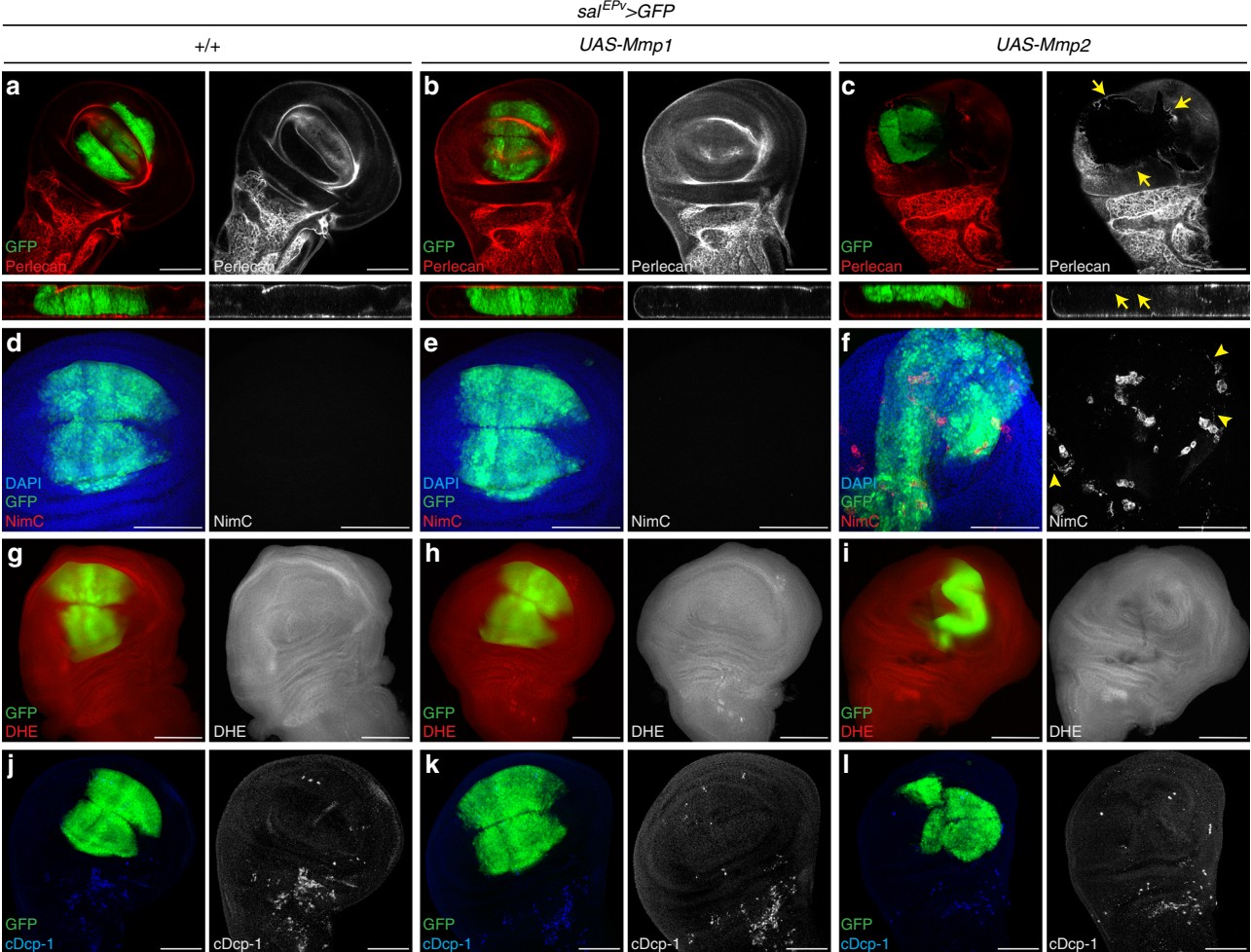

**Fig. 6 Mmp2-mediated BM damage recruits hemocytes.** GFP marks the *Sal^EPv* domain where Gal4 is expressed. Scale bars, 100 μm. **a–c** Overexpression of *Mmp2* is sufficient to damage the BM. Single slices focusing on the basal side of DP (top) and orthogonal (YZ) views (bottom) show big gaps in Perlecan labeling upon expression of *Mmp2* (yellow arrows) (**c**), indicating that BM is damaged. In contrast, the BM is intact with uniform Perlecan labeling upon expression of *Mmp1* (**b**) similar to control (**a**) indicating *Mmp1* does not damage BM. Quantified in Supplementary Fig. 14. **d–f** Damage to BM recruits hemocytes. Hemocytes are detected by NimC (red left; gray right). Hemocytes are usually absent from wing discs (control **d**). Expression of *Mmp1* does not recruit hemocytes to the discs (**e**). However, upon expression of *Mmp2* (**f**), high numbers of hemocytes are recruited to the discs which have damaged BM. A few of these recruited hemocytes show "activated" morphology and are indicated by yellow arrowheads. Quantified in Supplementary Fig. 14. **g–i** Expression of *Mmp2* does not cause ectopic production of ROS (**i**), similar to control (**g**) and expression of *Mmp1* (**h**). ROS is detected by dihydroethidium (DHE) dye (red left; gray right). Quantified in Supplementary Fig. 14. **j–l** Expression of *Mmp2* does not cause ectopic cell death indicated by absence of active caspases in (**l**), similar to control (**j**) and expression of *Mmp1* (**k**). Active caspases are detected by cDcp-1 antibody (blue left; gray right). Quantified in Supplementary Fig. 14.

neoplastic tumor models in *Drosophila*. Our data further suggest that *Mmp2* is upregulated in a ROS- and JNK-dependent manner. Mmp2 is responsible for BM damage of imaginal discs and the subsequent recruitment of hemocytes. Intriguingly, BM damage is observed specifically at the basal side of DP in undead discs, while the BM at the PE is unaffected. Consistently, Duox, the NADPH oxidase which synthesizes ROS in undead discs, is localized preferentially at the basal side of DP cells[49]. We also observed a strong accumulation of Mmp2 protein at the basal side of DP cells in undead discs. Combined, these observations strengthen the view that ROS and Mmp2 interdependently affect the integrity of the BM.

We are proposing the following model to explain the recruitment to and activation of hemocytes at undead and neoplastic eye-antennal imaginal discs.

In wild-type discs, naive hemocytes are present in tight cellular clusters around the morphogenetic furrow (Fig. 7a; Supplementary Fig. 1a,b). The function of the hemocytes at these discs

is unknown, but their morphology suggest that they are in a naive (inactive) state. In contrast, in undead discs, extracellular ROS produced by Duox, directly or indirectly, could provide an initial signal to activate the hemocytes present on the discs, causing them to secrete signaling factors, including the TNFα-ortholog Eiger[14], to activate JNK in undead cells (Fig. 7b, left). JNK in turn triggers the feedback amplification loop through induction of *hid*, but also of *Mmp2* (Fig. 7b, middle). Mmp2 protein accumulates at the basal side of the DP of undead discs, and cleaves and damages the BM, consequently recruiting additional hemocytes to the discs (Fig. 7b, right), which in turn release more signals for maintaining JNK activity. This further propels the amplification loop resulting in release of mitogens for AiP and overgrowth.

Although some of the hemocytes that are recruited to discs with damaged BM due to *Mmp2* overexpression display an activated morphology, they do not induce overgrowth (Fig. 6c, f). This is in stark contrast to hemocytes at undead discs, which are known to mediate AiP and overgrowth of undead tissue[14].

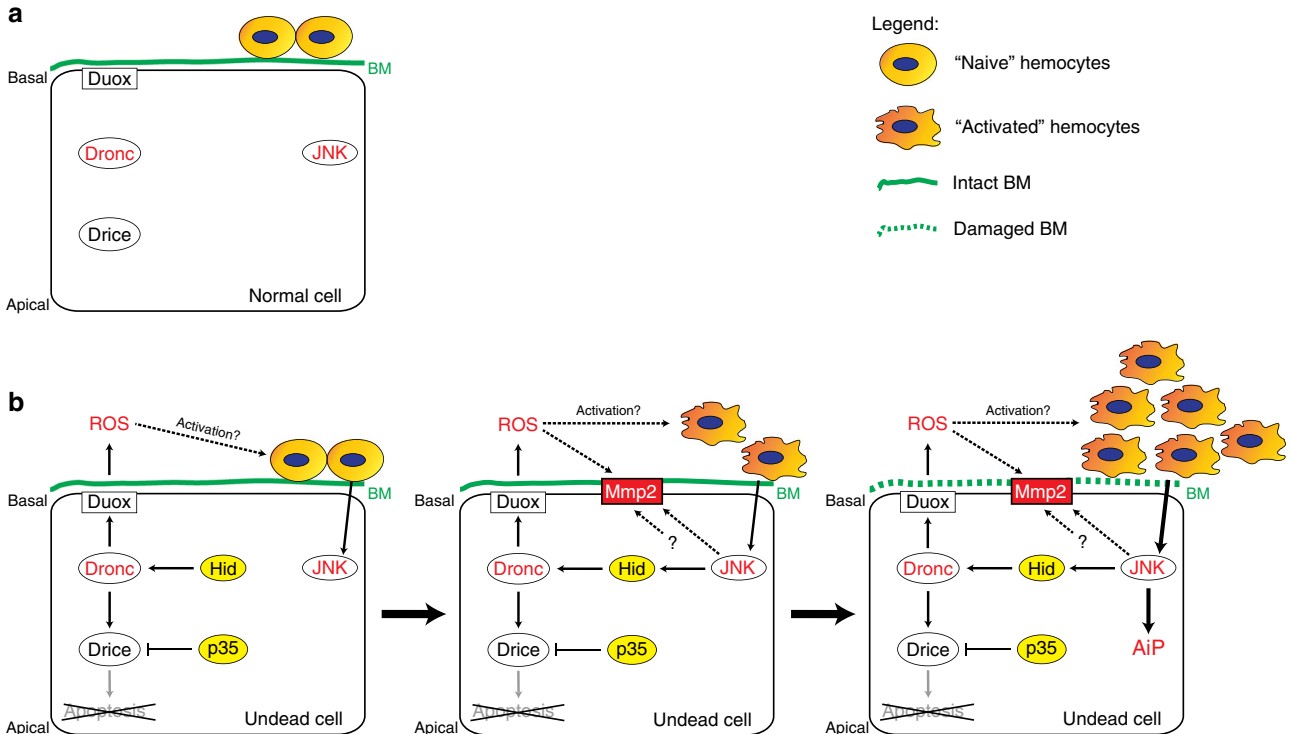

**Fig. 7 Summary model. a** In normal epithelial cells, Duox, JNK, Dronc, and Drice are synthesized, but are inactive. The basement membrane (BM) is intact and naive hemocytes are attached to the eye-antennal imaginal disc at the basal side of the disc proper. Please note the apical-basal polarity of the cells. **b** (left) In undead epithelial cells, expression of *hid* activates Dronc, but apoptosis is blocked due to co-expression of *p35* which inhibits Drice and generates the undead condition. Instead, Dronc activates Duox which synthesizes extracellular ROS. These ROS may provide the initial stimulus for activation of the naive hemocytes. **b** (middle) Activated hemocytes release factors including the TNF ortholog Eiger which signal back to the undead cells. JNK is activated and induces the expression of *hid*, setting up an amplification loop. JNK—directly or indirectly—also induces the expression of *Mmp2* resulting in accumulation of Mmp2 protein at the basal side of the plasma membrane. ROS may also further activate Mmp2. **b** (right) Mmp2 damages the BM at the basal side. The BM damage recruits additional hemocytes to further strengthen the amplification loop, resulting in AiP.

Therefore, hemocytes at undead discs behave differently than hemocytes under normal developmental or wound healing conditions[7]. It is enticing to speculate that ROS may provide a signal that causes the different behavior of recruited hemocytes at undead discs (Fig. 7b).

ROS may have more than one function in this process. Previous studies have proposed a ROS-mediated post-translational activation of mammalian Mmp2 via "cysteine-switch"[56]. A highly conserved cysteine in the amino-terminal region of Mmp2 directly ligands with the active-side Zinc, thereby maintaining Mmp2 proenzyme latency. Oxidation of this cysteine residue disrupts the interaction with Zinc leading to activation of the enzyme[56]. We showed that enzymatic activity of Mmp2 in undead discs is dependent on ROS. It will be interesting to determine if ROS would activate Mmp2 via the "cysteine-switch" mechanism in the undead model for damaging the BM. This would also explain why some areas have strong BM damage, others only weak damage, even though Mmp2 is uniformly upregulated at the basal side of DP throughout the undead disc. ROS are not produced uniformly in the undead disc, and discrepancies in ROS production would determine Mmp2 activation status across the disc. Upregulation and activation of MMPs is a common theme across multiple cancers[57], and it would be important to determine if a similar correlation exists between ROS production and MMP activation, and if the "cysteine-switch" mechanism is a common occurrence for ROS-dependent MMP activation in cancer.

Furthermore, the observation that hemocytes are recruited to imaginal discs in response to overexpressed *Mmp2* (Fig. 6f) in the

absence of apoptotic cells or detectable ROS (Fig. 6i, l) suggests that hemocyte recruitment per se is not dependent on apoptotic "find-me" signals or chemoattractants such as ROS. Instead, it may imply that circulating hemocytes are constantly surveying the microenvironment for damage, and once they encounter a disrupted BM, they are recruited for possible repair. An interesting follow-up question is to understand how the circulating hemocytes sense and adhere to the damaged BM. One possibility could be that BM damage exposes some otherwise cryptic receptors on the surface of the discs, which are not accessible to the hemocytes in discs with an intact BM. This may be a mechanism by which hemocytes distinguish a damaged versus healthy microenvironment.

Similar to *Drosophila* hemocytes, mammalian macrophages constantly survey the microenvironment for damage, tumorous growth or pathogens. In case of tumors, circulating monocytes are recruited to the tumor by chemokines secreted by the tumor cells, including apoptotic tumor cells[3–5,58]. Interestingly, apoptotic lymphoma cells also express and process mammalian Mmp2[3], showing clear parallels between vertebrates and our work in *Drosophila*. Future studies will determine whether macrophages are recruited to the tumor by a similar BM damage-sensing mechanism as seen for *Drosophila* hemocytes. Such a mechanism of macrophage recruitment would have an effect on therapeutic interventions that aim to block chemokine secretion by tumors, as they may not be sufficient to prevent macrophage accumulation.

*Mmp2* expression in undead cells is dependent on JNK signaling (Fig. 4h, i). That is surprising as previous work showed

that only *Mmp1*, but not *Mmp2*, expression is dependent on JNK in otherwise normal tissue[53] (Supplementary Fig. 10c). Even in the presence of $H_2O_2$, *Mmp2* was not upregulated in discs expressing an activated form of JNKK ($sal^{EPv} > hep^{CA}$) in the ex vivo experiments (Supplementary Fig. 10d), also explaining why we do not observe significant BM damage under these conditions (Fig. 3k, l). These results suggest that the undead environment triggered by co-expression of *hid* and *p35* provides a special condition for the upregulation of *Mmp2*.

BM degradation is the first step for metastatic tumor cells to leave their niche and invade other tissues[59–61]. However, BM damage is not sufficient to allow undead cells to leave their niche and invade other tissues. This suggests that additional signals are required for gaining metastatic properties along with BM damage. Future studies could explore possible mechanisms for inducing metastatic behavior of undead cells.

In conclusion, we described a mechanism by which ROS can dictate the recruitment of macrophages by modulating Mmp2 levels and activity that cause damage to the BM. Furthermore, these results indicate that undead tissue shares similar properties to neoplastic tumors with many overlapping "hallmarks of cancer". Overall, our data strongly imply the role of the tumor microenvironment in recruiting macrophages to the tumor, and also highlight the importance of *Drosophila* undead tissue as a model to study the interplay between tumors and immune cells.

## Methods

**Fly stocks**. The following mutant and transgenic stocks were used: *ey-Gal4*, *sal^{EPv}-Gal4* (ref. [62]), *UAS-p35*, *UAS-hid*, *UAS-hep^{CA}*, *UAS-Ras^{V12}*, *hmlΔnRFP* (ref. [63]), *lgl^4* (ref. [64]), *vps25^{N55}* (ref. [40]), *hpo^{42–47}* (ref. [65]), *srp^{neo45}* (ref. [51]). UAS-based RNAi and overexpression lines for the following genes were obtained from the Bloomington *Drosophila* Stock Centre: *Mmp1* (#31489, #58700, #58701, #58702, #58703), *Mmp2* (#31371, #58705, #58706, #61309, #65935), *RFP* RNAi (mock RNAi) (#67984), *UAS-JNK^{DN}* (*Bsk^{DN}*) (#6409). *UAS-Duox* RNAi and *UAS-hCatS* were a kind gift from Won-Jae Lee (ref. [66]). The exact genotype of the following stocks is

> *ey > p35*: *ey-Gal4 UAS-p35/CyO*
> *ey > hid,p35*: *UAS-hid; ey-Gal4 UAS-p35/CyO,tub-GAL80*
> *eyeful*: *ey-Gal4 > UAS-delta, GS88A8 UAS-lola and UAS-pipsqueak* (ref. [43])

The *ey > hid,p35* stock was crossed to *w1118* as control or RNAi and mutant lines, and the offspring was scored for suppression of head capsule overgrowth phenotype. Screening results are presented as percentage of animals with either wild-type or overgrown phenotype.

All crosses were maintained at 25 °C. Except where noted, most crosses were performed using the *Gal4/UAS* system. Tumor mosaic eye discs were induced using the MARCM technique[67] with *ey-FLP* to generate mosaics with tumor clones marked by GFP[38].

**Immunofluorescence staining and visualization**. Imaginal discs were dissected from late 3rd instar larvae in 1× PBS, fixed with 4% paraformaldehyde (PFA) for 30 min at room temperature (RT), washed with 1× PBS with 0.3% Triton X-100, blocked with Normal Donkey Serum, and stained with primary antibodies overnight at 4 °C. The following primary antibodies were used: rabbit anti-Perlecan (1:1000, kind gift from Stefan Baumgartner)[68], rabbit anti-Laminin (1:500, abcam ab47651), mouse anti-NimC (1:100, P1a,P1b, kind gift from István Andó)[69], rabbit anti-Mmp2 (1:500, #436, kind gift from Kendal Broadie)[70], rabbit anti-cleaved Dcp-1 (1:250, Cell Signaling), rabbit anti-phospho JNK (1:200, Promega), rat anti-ELAV (1:50), and mouse anti-Mmp1 (1:50) (DSHB). Secondary antibodies were donkey Fab fragments from Jackson ImmunoResearch (used at 1:600 each and incubated at RT for 3 h). Eye-antennal discs were counter-labeled with the nuclear dye DAPI to visualize tissue outline.

DHE labeling was performed on unfixed imaginal discs as follows—Imaginal discs were dissected from 3rd instar larvae in fresh Schneider's medium, incubated in DHE (final concentration 30 μM; Invitrogen, Cat#D23107) solution for 5 min, washed three times with PBS, mounted in Vectashield mounting media, and imaged immediately.

Images were obtained with a Zeiss LSM 700 confocal microscope, analyzed with Zen 2012 imaging software (Carl Zeiss) and processed with Adobe Photoshop CS6.

**Electron microscopy**. Samples were processed and analyzed at the University of Massachusetts Medical School Electron Microscopy core facility.

**Scanning electron microscopy**. Imaginal discs from 3rd instar larvae were dissected and fixed in 2.5% (v/v) glutaraldehyde in 0.1 M sodium cacodylate buffer

(pH 7.2) for 1 h at RT, and then at 4 °C overnight and post-fixed in 1% (w/v) osmium tetroxide for 1 h at RT. The imaginal discs were dehydrated through a graded series of ethanol to 100% and then transferred into porous sample holders and dried at a critical point in liquid $CO_2$. The dried samples were transferred to aluminum SEM stubs coated with adhesive carbon tape, the edges were painted with silver conductive paste and the SEM stubs were sputter-coated with Au/Pd (80/20). The specimens were then examined using an FEI Quanta 200 FEG MKII scanning electron microscope at 10 kV accelerating voltage.

**Transmission electron microscopy**. Imaginal discs from 3rd instar larvae were dissected and fixed in 2.5% glutaraldehyde in 0.1 M sodium cacodylate buffer pH 7.2 for 1 h at RT and post-fixed with 1% osmium tetroxide for 1 h at RT. Samples were in block stained with a 1% uranyl acetate aqueous solution (w/v) at 6 °C overnight, dehydrated in ethanol, and infiltrated first with two changes of 100% propylene oxide and then with a 50%/50% propylene oxide/SPI-Pon812 resin mixture, and finally polymerized at 68 °C in flat pre-filled embedding molds. The samples were then reoriented, and ultra-thin sections (~70 nm) were placed on copper support grids and contrasted with lead citrate and uranyl acetate. Sections were examined using the CM10 TEM with 100 kV accelerating voltage, and images were captured using a Gatan TEM CCD camera.

**Ex vivo $H_2O_2$ experiment**. Imaginal discs were dissected from 3rd instar larvae in fresh Schneider's medium. The medium was then replaced with 50 μM $H_2O_2$ (Sigma-Aldrich, Cat#H1009) solution in Schneider's medium or fresh Schneider's medium (control) and incubated for 60 min at 25 °C. Eye discs were washed with PBS and fixed in 4% PFA in PBS for 30 min at RT, followed by immuno-fluorescence and imaging as described above.

**Counts and analysis of on-disc hemocytes**. Imaginal discs were dissected from 3rd instar larvae from control and undead animals fixed with PFA and stained using anti-NimC antibody and DAPI. Imaginal discs were imaged using confocal microscopy. Total number of hemocytes attached to the discs were counted using multi-point tool in ImageJ software. Activated hemocytes were identified by presence of one or more cellular extensions detected by NimC labeling and counted by the multi-point tool in ImageJ.

**Mmp2 activity assays**. Mmp2 fluorogenic activity assays were performed using the fluorescein labeled DQ-gelatin, as part of the EnzChek™ Gelatinase/Collagenase Assay Kit (Invitrogen E12055), following manufacturers' instructions using 25 μg/ml substrate. Imaginal discs were dissected from 3rd instar larvae and lysed with 100 μl of lysis buffer (50 mM TrisHCl, pH 7.4, 150 mM NaCl, 1% Triton X-100, and Halt protease inhibitors (Thermo-Fisher)). Samples were sonicated for 5 min using BioRuptor according to the manufacturer's protocol (medium power, 10 min, 30-s on and 30-s off at 4 °C), and protein concentrations of supernatants were measured by Bradford Assay. Protein lysates were added directly to the substrate and incubated at 37 °C for different time points, after which the fluorescence intensity was measured on Beckman Coulter DTX 880 microtiter plate reader at 450/8 nm excitation and 535/25 nm emission.

The in situ zymography was performed as previously described with minor modifications[54]. Mmp2 activity was detected using the fluorescein labeled DQ-gelatin substrate on dissected unfixed imaginal discs. Imaginal discs were dissected from 3rd instar larvae in fresh Schneider's medium. The medium was then replaced with 10 μg/ml DQ-gelatin substrate in Schneider's medium and incubated in the dark for 15 min at 29 °C, washed with PBS, mounted in Vectashield mounting media, and imaged immediately.

For gel zymography assays, imaginal disc protein lysates were prepared exactly as described for the fluorogenic assay. Thirty micrograms of protein lysate was resuspended in 2x Novex non-reducing sample buffer and samples were run on Novex 10% Zymogram Plus gel containing 0.1% gelatin (ZY00100BOX). After electrophoresis, gels were incubated with Novex Zymogram Renaturing Buffer and Novex Zymogram Developing Buffer, then stained with SimplyBlue Safestain (LC6060), all according to manufacturer's instructions (Invitrogen).

**Quantitative real-time PCR (qRT-PCR)**. Total RNA from 20 eye discs was isolated using Trizol (Invitrogen). cDNA was synthesized from 1 μg of RNA with the QuantiTect Reverse Transcription Kit (Qiagen). For RT-PCR analysis, 50 ng of cDNA per reaction was subjected to 40 amplification cycles. Real-time quantification was performed using Power SYBR green PCR Master Mix reagents (Applied Biosystems) on QuantStudio 6 Flex Real-Time PCR System (Applied Biosystems) according to the manufacturer's instructions. *rp49* expression was used as an internal control for normalization. Three experiments for each genotype were averaged. Sequences for isoform-specific primers are as follows:

> Rp49 forward: CCAGTCGGATCGATATGCT
> Rp49 reverse: ACGTTGTGCACCAGGAACT
> Mmp1 forward: CACCTACAAGAACGGCAAGA
> Mmp1 reverse: CGCTGATTTCCTTAGGGTAGAC
> Mmp2 forward: TTACCTCATGCAGTTTGATTATCT
> Mmp2 reverse: CTCCACGTAAGATCCGTTCTG

**Quantification, statistical analysis, and reproducibility**. For quantification of confocal images, the 'Histo' function of Zen 2012 imaging software (Carl Zeiss) was used. Region of interest (entire eye-antennal imaginal discs or $sal^{EPv}$-Gal4 domain of wing imaginal discs) was outlined for each disc and mean fluorescence signal intensity was determined. For Perlecan, Laminin, and Mmp2 staining intensity quantification, single sections at basal side of DP and PE were selected and separately measured. For Perlecan and Laminin quantifications, because the BM was only affected at the DP, but not at the PE (Supplementary Fig. 3), the signal intensity at PE was used as an internal control to normalize the signal intensity at DP for each disc, and graphed as "Mean Normalized Fluorescence at DP to PE". For DHE, pJNK, and cDcp-1, staining intensity was measured for the maximum intensity projections. At least three biological repeats were performed for all experiments. Total number of discs used for quantification are indicated in the legends for each figure. Analysis and graph generation was done using GraphPad Prism 7.04. Statistical analysis for Perlecan, Laminin, Mmp2, DHE, pJNK, and cDcp-1 intensity was performed using one-way ANOVA with Holm–Sidak test for multiple comparisons. Data are represented as the mean ± standard error of the mean (SEM) of pooled results obtained from the indicated number of samples in each experiment. Statistical analysis for hemocyte counts was performed using one-way ANOVA with Tukey's test for multiple comparisons. Statistical analysis of $Mmp1$ and $Mmp2$ qPCR was performed using two-tailed unpaired Student's $T$ test. Data are represented as mean ± standard deviation (SD). Statistical analysis for adult counts was performed using two-tailed Fisher's exact test. Level of significance are depicted by asterisks in the figures: $*p < 0.05$; $**p < 0.01$; $***p < 0.001$; $****p < 0.0001$.

**Reporting summary**. Further information on research design is available in the Nature Research Reporting Summary linked to this article.

## Data availability

The source data underlying Figs. 1, 3, 4, 5 and Supplementary Figs. 1–14 are provided in the Source Data file. Other data that support the study are available from the corresponding author upon request.

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

## Acknowledgements

We would like to acknowledge Lara Strittmatter and the UMMS Electron Microscopy Core for helping us with the SEM and TEM. We would like to thank István Andó, Stefan Baumgartner, Kendal Broadie, Maria Dominguez, Won-Jae Lee, Andrea Page-McCaw, Duojia Pan, Jose Carlos Pastor-Pareja, Jianjun Sun, Lei Xue, the *Drosophila* stock center in Bloomington (BDSC) and the Developmental Studies Hybridoma Bank (DSHB) for antibodies and fly stocks. This work was funded by the National Institute of General Medical Science (NIGMS) under award number R35 GM118330. The EM work was supported by the National Center for Research Resources under award number S10 OD025113-01 (TEM), S10RR021043 (SEM), and SI0OD021580 (Ultramicrotomy). The content is solely the responsibility of the authors and does not necessarily represent the official views of the NIH.

## Author contributions

N.D. performed all experiments. N.D. and A.B. designed the experiments and wrote the paper. A.B. secured funding for the project.

## Competing interests

The authors declare no competing interests.
