## [Peer Review File · Nature Communications]

Reviewers' Comments:

Reviewer #1:

Remarks to the Author:

The manuscript by Diwanji & Bermann addresses the question of how macrophages are recruited to tumors, an observation that holds in *Drosophila* as well as in human tumors. The main tumor model used is the "undead" eye-antennal disc in the *Drosophila* larva, in which cells are induced to initiate apoptosis by expression of *hid* but are inhibited from completing the process by expression of *p35*. In these discs, apoptosis-induced proliferation signals continue, resulting in over-proliferation and tumor formation. The authors report that in undead discs, the basement membrane is damaged, as imaged by anti-perlecan staining. They survey the phenomenon of basement membrane damage in fly tumor models, analyzing four additional tumor models, two neoplastic and two hyperplastic, and they find that basement membrane damage is associated with the neoplastic but not the hyperplastic models. Further, they note that although all tumor models recruit hemocytes, in the neoplastic models, ROS levels are elevated, JNK is phosphorylated, and hemocytes are activated, thus establishing a correlation between these phenomena. Investigating how basement membranes become damaged, the authors find that ROS and JNK are required but are not sufficient for BM damage. Additionally, they find both fly MMPs are upregulated in undead discs. Importantly, *Mmp2* is both required and sufficient to cause basement membrane damage. Finally, they overexpress *Mmp2* and find it sufficient to recruit hemocytes, which they interpret as basement membrane damage is sufficient to recruit hemocytes.

Although the model put forth in the paper is clear, unfortunately many of the claims are not sufficiently supported by the data, and the mechanisms underlying their model are not explored. The assay showing basement membrane damage is not convincing. Many interpretations rest on qualitative differences that are not clearly defined or quantified and shown in a single image. Additional models may explain their results. These concerns are described below in detail.

1. The focus of the paper is on basement membrane damage, but this data must be stronger. What is damage? Anti-perlecan staining is the only assay used to evaluate the basement membrane, and the data is over-interpreted. To assay basement membrane (rather than perlecan itself), other assays must be employed, such as electron microscopy, super-resolution microscopy, mechanical testing (eg, AFM), and/or fluorescence of other components such as collagen IV (Vkg, Cg25C) or laminin (beta1 is the easiest). In *Drosophila*, Perlecan was recently identified as an MMP substrate by Wu-Min Deng's group, and so it is a distinct possibility that the effect may be mediated directly by perlecan and its effect on signaling, rather than the basement membrane.
2. The perlecan data itself is not convincing: single examples of gaps in perlecan are shown for each genotype in z-section (Figs 1D", S4C", 2A", 2C", 3A"), yet similar gaps are present in many samples they say are undamaged (Fig 1C", S2B" S4A", 3B"). How exactly is a gap defined? Either a quantitative definition must be applied uniformly, or samples must be scored blind by someone who cannot discern their identities from the phenotype; in either case, the frequency of damage must be reported. Analyzing mean Perlecan fluorescence levels is not a substitute, as damage is inferred from the presence of gaps and discontinuities, not total fluorescence. (Levels are also complicated by antibody staining: if samples were permeabilized before staining, intracellular and mis-localized perlecan are included in this metric – and it looks to me as if one feature of neoplastic discs may be the presence of apical or intracellular perlecan.)
3. Several other findings lack quantification. To make the observation that hemocytes are activated in some cases but not others, there must be quantification of the activated phenotype (Fig. S3). Same for ROS production in Fig. 2.
4. It is striking that when *Duox* is knocked down or catalase is overexpressed (Fig. 3A-C) the disc morphology is so much improved. Fig 3A looks like it is overly compressed, which leads me to speculate about an indirect model for how ROS lead to *Mmp2* production: High levels of H₂O₂ would be expected to increase local peroxidase-mediated basement membrane (col4) crosslinking (Bhave et al, 2012; McCall et al, 2014), which would increase basement membrane stiffness (Bhave et al 2017). Stiffness would also be increased by the reduced levels of perlecan (Pastor-

Pareja et al, 2011), likely because perlecan is no longer targeted basally (stemming from problems with the apical-basal polarity of the undead cells). In the undead disc, cells are over-proliferating, but constraint by an overly stiff basement membrane could create a JNK-mediated stress signal, triggering the expression of basement membrane cleaving enzymes like Mmp2 to reduce stiffness, a homeostatic circuit that would normally restore the balance between cells and basement membrane. In this case, Mmp2-cleavage of the basement membrane releases stiffness and gives cells room to grow within the basement membrane; knockdown of Mmp2 inhibits growth. High levels of Mmp2 then recruit hemocytes, which the authors acknowledge has been shown previously.

Of course this model may be incorrect – especially if the basement membrane is not damaged and only perlecan is affected. However, it explains many of the authors observations, and makes stiffness of basement membrane a central question. Matrix stiffness is a central feature of human tumors so it may be worth exploring.

5. Although the model put forth by the authors is conceptually clear, many steps of the mechanism are still unsupported, and without at least some of these mechanistic insights, the paper is less impactful. Examples are, how does ROS upregulate Mmp2? Can the authors supply evidence (eg Western blot, zymogram) that the cysteine switch of Mmp2 is activated by ROS? How does JNK upregulate Mmp2 in the undead context but not others? How do hemocytes become activated by ROS? How does Perlecan cleavage/basement membrane cleavage attract hemocytes?

6. The distinction between the neoplastic and hyperplastic tumor models would be stronger if it the authors mentioned the criteria underlying this distinction.

7. A couple of reference issues: Reference 46 (Llano et al, 2000) has been retracted so probably shouldn't be cited. It was recently established that both Mmp1 and Mmp2 have membrane tethered and secreted forms (LaFever et al, 2017), so lines 103-4 can be altered to incorporate this new information.

Reviewer #2:

Remarks to the Author:

Summary and assessment of novelty and interest:

The work by Diwanji and Bergmann studies the recruitment of macrophages to the undead model for apoptosis induced proliferation in larval imaginal discs.

They claim to have identified that

- 1) ROS and BM damage occur in neoplastic models.
- 2) ROS is required but not sufficient for BM damage
- 3) ROS and JNK are needed to upregulate MMP2 in the undead model
- 4) MMP2 damages the BM leading to macrophage recruitment

Some of the findings in their details are new, (although Suppl Fig S5 D-F were shown already in the Bergmann lab's 2016 Curr Bio paper in Fig 2G-I (Fogarty et al, PMID 26898463)) but the overarching conceptual framework is not very novel. And Fig 2B was already shown in Fig 1E of Perez et al 2017, PMID 28853394

ROS induction in the undead model is known (Fig 1 A-K Fogarty et al). That BM damage occurs has already been shown in a neoplastic model (Fig 4 Pagliarini and Xu, 2003 PMID 14551319, Fig 1 Pastor-Pareja et al DMM 2008, PMID 19048077). The role for ROS in recruiting/ activating hemocytes in the undead model has been shown (Fig 2, Fogarty et al). ROS is already known to induce a different MMP, MMP1 (Fig 3, Fogarty et al). That JNK can induce this same other MMP, MMP1, is known (Fig 2.C,H Uhlrova et al 2006 PMID 170827730). That MMP2 can induce BM damage has been shown (Fig 5 Pastor-Pareja et al DMM 2008, PMID 19048077)). That such BM damage by MMP2 correlates with macrophage recruitment has already been shown in a neoplastic model (Fig 1, Pastor-Pareja et al). That damage to the BM by MMP2 is sufficient to lead to

macrophage recruitment to the wing disc or salivary gland has already been shown (Fig 5 Pastor-Pareja et al).

This submitted paper adds new findings showing the necessity of MMP2 for hemocyte recruitment in this undead model, and data showing that its induction depends on both ROS and JNK signaling. This, I would predict, will be interesting to the field and expand their understanding but will not change current models greatly.

Suggested experiments to extend their finding:

They could strengthen their main new conclusion by testing if either ROS treatment or JNK signaling on its own can induce MMP2 expression, and if not, showing the sufficiency of H₂O₂ treatment plus JNK activation to induce MMP2 and hemocyte recruitment. They show in this paper that neither of these treatments on their own can induce basement membrane damage (Fig 3E-F, Fig S4B), however since their data shows that where MMP2 is induced is not always where BM damage occurs (see discussion lines 354-369), they should examine MMP2 expression itself. This is not much work as they already have the line (salEPV>hepCA, GFP), and just need to dissect discs and dunk some of them also into H₂O₂.

Major problems with data:

I am not convinced by the data presented in some places due to a lack of quantitation and/or a discordance between the quantitation presented and the supporting immunofluorescent panels.

Perlecan data:

Most strikingly, the data provided analysing basement membrane damage or lack thereof do not clearly match the conclusions drawn or the quantitation made (when quantitation is provided, which is not always the case). In Figure 1 for example, it is hard to reconcile the staining shown for Perlecan in B'-D' or B''-D'' with the quantitation shown in E. They just don't seem to match at all; the changes in B''-D'', particularly between C'' and D'' are very subtle and not clearly evident, not matching the quantitation shown in E. Looking at B-D' one would conclude that there is a reduction in Perlecan levels already in the ey>p35 control, again not matching what is in the quantitation. I have similar issues for Figs 3A-C'' where the changes in the DP shown in the quantification are not obvious to me and where the Perlecan staining in the PE in panel C'' looks completely absent compared to A'' while being at the same level in the quantification. Same issues for Fig 5 A-C''.

This lack of matching between quantitation and images for Perlecan is particularly striking when one compares it to the data for Mmp2 shown in Figure 4, where the again two fold differences in levels in the DP side shown in the quantitation in Fig 4D are completely obvious and correlate exactly with what is shown in panels 4A-C and A''-C''.

I tend to believe the data in quantitation of multiple samples over a single example, however if there is no single set of examples that roughly matches the average of the quantitation, that is strange.

There is also no comprehensive description of how the Perlecan (or MMP2) quantitation was conducted, to make me sure that difference between the data shown and quantitation isn't due to some problem with the quantitation. Was the quantitation done in the region of overgrowth? Were equivalently sized regions were outlined from a single z stack focusing on the basal side from the different samples and had their level of fluorescence assessed? Was there some internal control to normalize for differences in general staining between samples by assessing the level of Perlecan in a region where no change would be expected (ie a part of the disc that isn't undead). If that isn't appropriate because the changes appear to extend beyond the region of the hid p35 expression can one use the staining for some other ubiquitous protein as a control for differences in fixation that might alter antibody staining generally?

Request 1, additional production of extant data and textual change:

Please provide more of the data that was quantitated to clarify this issue for me, and/or provide me with a convincing explanation for what is going on. Please improve the description of Perlecan quantitation in the methods.

Let's assume that the quantitation is fine and there is some reason why the pictures shown don't match it very obviously. Then, given that the pictures are not easy to interpret, having Figures that provide only one picture of Perlecan without quantitation is also problematic.

Request 2, additional analysis and textual change, Fig 3, S2 and S4:

Please quantitate the level of Perlecan in the H₂O₂ treated case (corresponding to what is shown in Fig 3E, F), to be sure there is no change.

Please quantitate Perlecan levels for Figure S2 A",B', where the pictures shown do not at all support the conclusion stated.

Please quantitate the level of Perlecan for Fig S4 A-C to anchor the conclusion that JNK is required for BM damage and include the control for the saEPV panel S4B.

For all this quantitation for these panels list how many discs were examined and do the statistics.

Request 2: Further analysis, Fig 2

I didn't find Figure 2 on the correlation of BM and ROS very convincing, and found this figure mostly useful for assessing if overgrowth on its own can cause BM damage (which I think would make it a better supplemental Figure). But in any case they need to quantitate whatever they show (BM damage and ROS if both are still included).

Request 3, Further analysis and textual change related to MMP2 data in Fig 4:

a) It appears as if there is some upregulation of MMP2 in Fig 4B (in the ey>p35 control) based on the whole disc view, even though this is not seen in the close up in B'. Please clarify. This might fit given that it is known that there is already hemocyte recruitment in the ey>p35 case (see below, Hemocyte point c).

a) In Fig 4C they see induction of MMP2 throughout the entire disc, not just the portion that is undead. This does not fit their model as I understand it in which upregulation of JNK signaling in the undead cell (Fig 7) leads to higher levels of MMP2 in those undead cells. They explain in the discussion how localized ROS could potentiate MMP2 activity to allow the localized presence of BM damage despite the uniform MMP2 induction. However they don't explain how localized JNK and localized ROS induction lead to ubiquitous MMP2 expression. There could be some other component downstream of both ROS and JNK that diffuses broadly to induce MMP2 expression in the whole disc (unless something is wrong with their staining). To make sure the MMP2 Ab isn't binding nonspecifically in undead discs, they should do a control staining in the background in which the undead disc also expresses RNAi against MMP2 (the line they already have, given data in Fig 5C). If the antibody is truly binding MMP2 then they should see downregulation of MMP2 Ab staining in the undead region of the disc. If this control is fine and if they agree with my interpretation of their results as described above, modify Fig 7 accordingly.

b) In Fig 4 D they only quantify MMP2 in the DP. It appears as if MMP2 is upregulated in DP and PE based on the picture in 4C. To assess this they should also quantify expression in the PE.

Request 4, further analysis hemocyte data:

a) Fig 5E-G Please quantitate the amount of hemocyte attachment. It appears as if there is a reduction in the Mmp1 RNAi case as well based on the panel shown.

b) There is no clear data to substantiate the claim that hemocytes are inactive in the hyperplastic model shown in S3 or in Fig S4C when there are fewer hemocytes present. Please show a close up to show lack or presence of extensions and quantitate their extent.

c) in this paper there is no examination and quantitation of the *ey>p35* control for hemocyte recruitment, only *ey-GAL4* compared to *ey>p35 >Hid* (Fig S1). The Fogarty et al paper Fig 3A showed that hemocytes are already recruited in *ey>p35* but that in *ey>p35>hid* hemocytes shift to an activated state.

This raises the question as to whether MMP2 is required also for recruitment of hemocytes in the *ey>p35* control. They should analyse this. If yes, then this MMP2 pathway might not be particularly specific for recruitment in the undead model but more generally induced in response to blocking apoptosis. This might also fit with the data shown in Fig 1C, C" where it appears as if there is a downregulation of Perlecan in this control. If this is the case they would need to rewrite the paper to focus on macrophage recruitment to cells in which apoptosis is blocked and extend the dissection of MMP2 induction in the *ey>p35* system. They need to shift the framing, which is currently about understanding recruitment to a tumor model.

Request 5: Further analysis, Fig 6

Please quantitate the DHE levels and the DCp-1 staining for all genotypes.
List n and do statistics.

Textual changes:

1) In the introduction in line 53, they don't mention that MMP2s have been shown to be sufficient to recruit hemocytes (shown in Pastor-Pareja et al, their ref 12, which is only mentioned glancingly as showing hemocyte recruitment to tumors.)

2) line 55, I wouldn't say "mimicking", I would say "analogous to"

3) In introduction lines 70-73 they reference previous findings on ROS and JNK driving AiP but don't mention the previous data showing that ROS is required for hemocyte recruitment in Fig 3E, F Fogarty et al. They should.

4) In Fig 2, there is little evidence that ROS and BM alterations overlap, either with the region of the tumor (in the RasV12 case) or with each other (See Fig 2B where there are strong DHE clusters both far away from or neighboring tumor clones as well as some within clones.) Thus when saying (line 167) that DH puncta are observed they should add "in and around tumor clones". I would also qualify their statements about correlation (line 171) to say, "the appearance of BM damage and ROS in the disc seems to occur coincidentally upon the induction of neoplastic growth. Since they have not assessed both ROS and BM damage in their AiP model, they should strike lines 171-173.

5) line 220, should be S4B,B" not S4D,D"

6) line 233, and 973. They can't conclude in Fig S4 that hemocytes are not necessary for damaging the BM if hemocytes are still present, which they demonstrate. Please soften the conclusion to saying damage to the BM is not linearly correlated with the amount of hemocytes (if the quantification shows equivalent levels of BM in the *ey>hid* and the *ey>hid srpneo45/+* situation).

6) Text line 846-7, I think they mean "discs still had gaps as in the mock"

7) They explain why they shift to the wing disc model for Fig 6 in the text, but they do not explain why they used this alternate model for the data shown in Fig S2 and Fig S4. Please do so.

8) They should list in the methods the exact genotype of the stocks used in all Figure panels

Reviewer #3:

Remarks to the Author:

This is a fascinating study from Diwanji and Bergmann that investigates the mechanism by which *Drosophila* hemocytes are recruited to undead cells to trigger a signalling loop that ultimately leads to tissue overgrowth via AiP. Using the powerful genetics available in the fly the study uncovers a mechanism whereby ROS production together with JNK signalling within the undead cells leads to the activation and accumulation of MMP2 which in turn leads to a loss in basement membrane integrity. They also show evidence that it is this basement membrane damage that is responsible for the recruitment of hemocytes to the undead cells which play a key role in the signalling loop. The signalling they uncover using the undead – AiP model appears to hold true in neoplastic tumours. This paper is well written, the experiments are well designed and the data of good quality. The hemocyte recruitment and 'activated morphology' need to be quantified throughout the paper but if the authors can address that together with the following points then I think it would be an excellent paper for Nature Communications.

1. The authors refer to hemocytes at tumours and undead cells as having undergone 'alternative activation' which provides them with a TAM-like activity. I think that is a stretch too far. There is little evidence of these cells really behaving as TAMs and their 'activated morphology' is at present not convincing and needs more rigorous quantification (see below).

2. Supp Figure 1 A'' apparently shows a high magnification zoom of A' but the sample looks completely different. The same is true of B' and B''. The authors need to highlight the area being magnified on the lower mag image. For both these images as well as Figure 5F the 'activated morphology' of hemocytes is not clear. The authors should show higher magnification images to clearly demonstrate the protrusions on activated hemocytes and quantify this protrusive morphology as well as hemocyte recruitment (hemocyte number) throughout the paper. The current images are not very convincing/clear

3. The authors show that providing either ROS or JNK signalling alone is insufficient to cause basement membrane damage and therefore conclude that both are required. They should directly show this using their *ex vivo* assay by exposing discs that express hepCA to hydrogen peroxide and quantifying basement membrane disruption

4. The authors use UAS-MMP2 overexpression to generate basement membrane damage in the absence of ROS and still get hemocyte recruitment. What happens if they generate ROS from undead cells but block basement membrane damage? They have in fact already done this in the experiment described in Figure S4 (i.e JNK-DN *ey>hid p35*). Are hemocytes recruited to the undead cells in this experiment?

5. The title doesn't seem to fit the take home message. The basement membrane damage requires both ROS and JNK signalling so the title should either include JNK as well as ROS or they should lose ROS to simply state 'MMP2-mediated basement membrane damage recruits macrophages to overgrown tissue in *drosophila*'

Minor points

1. Why do the authors use the wing disc for the experiment in Figure S2 rather than the eye disc as is used for the rest of the paper?

2. line 378. shouldn't 'cryptic receptors' read 'cryptic ligands'?

3. The authors should cite the following paper in their discussion that demonstrated that upon induction of apoptosis, lymphoma cells express and process MMP2 showing clear parallels between

vertebrate studies and the data the authors present here in the fly: Ford et al, 2015, Current Biology 25(5):577-88

We would like to thank the reviewers to take valuable time to review the manuscript and for their insightful comments and suggestions to further improve it. We are happy to say that we have addressed all concerns of the reviewers. We have added more data to the main figures (especially in Figs 1, 3 and 4) and added seven new supplementary figures. We think that the manuscript has significantly improved. Please see the detailed point-by-point response below.

Reviewer #1 (Remarks to the Author):

The manuscript by Diwanji & Bergmann addresses the question of how macrophages are recruited to tumors, an observation that holds in *Drosophila* as well as in human tumors. The main tumor model used is the “undead” eye-antennal disc in the *Drosophila* larva, in which cells are induced to initiate apoptosis by expression of *hid* but are inhibited from completing the process by expression of *p35*. In these discs, apoptosis-induced proliferation signals continue, resulting in over-proliferation and tumor formation. The authors report that in undead discs, the basement membrane is damaged, as imaged by anti-perlecan staining. They survey the phenomenon of basement membrane damage in fly tumor models, analyzing four additional tumor models, two neoplastic and two hyperplastic, and they find that basement membrane damage is associated with the neoplastic but not the hyperplastic models. Further, they note that although all tumor models recruit hemocytes, in the neoplastic models, ROS levels are elevated, JNK is phosphorylated, and hemocytes are activated, thus establishing a correlation between these phenomena. Investigating how basement membranes become damaged, the authors find that ROS and JNK are required but are not sufficient for BM damage. Additionally, they find both fly MMPs are upregulated in undead discs. Importantly, *Mmp2* is both required and sufficient to cause basement membrane damage. Finally, they overexpress *Mmp2* and find it sufficient to recruit hemocytes, which they interpret as basement membrane damage is sufficient to recruit hemocytes. Although the model put forth in the paper is clear, unfortunately many of the claims are not sufficiently supported by the data, and the mechanisms underlying their model are not explored. The assay showing basement membrane damage is not convincing. Many interpretations rest on qualitative differences that are not clearly defined or quantified and shown in a single image. Additional models may explain their results. These concerns are described below in detail.

1. The focus of the paper is on basement membrane damage, but this data must be stronger. What is damage? Anti-perlecan staining is the only assay used to evaluate the basement membrane, and the data is over-interpreted. To assay basement membrane (rather than perlecan itself), other assays must be employed, such as electron microscopy, super-resolution microscopy, mechanical testing (eg, AFM), and/or fluorescence of other components such as collagen IV (Vkg, Cg25C) or laminin (beta1 is the easiest). In *Drosophila*, Perlecan was recently identified as an MMP substrate by Wu-Min Deng’s group, and so it is a distinct possibility that the effect may be mediated directly by perlecan and its effect on signaling, rather than the basement membrane.

RESPONSE: We followed the suggestions of the reviewer and have performed some of the requested experiments. We examined undead eye discs using an antibody against Laminin and confirmed the data obtained with the Perlecan antibody demonstrating damage to the basement membrane (BM). We also performed scanning and transmission electron microscopy (SEM and TEM) and observed damage at the surface of the undead discs, but not control discs. In all these cases, the damage to the BM is restricted

to the disc proper side. The BM at the peripodial epithelium is still intact. These data are presented in the revised Fig. 1 and in Suppl. Figs S2 and S3.

2. The perlecan data itself is not convincing: single examples of gaps in perlecan are shown for each genotype in z-section (Figs 1D", S4C", 2A"', 2C"', 3A"), yet similar gaps are present in many samples they say are undamaged (Fig 1C", S2B" S4A", 3B"). How exactly is a gap defined? Either a quantitative definition must be applied uniformly, or samples must be scored blind by someone who cannot discern their identities from the phenotype; in either case, the frequency of damage must be reported. Analyzing mean Perlecan fluorescence levels is not a substitute, as damage is inferred from the presence of gaps and discontinuities, not total fluorescence. (Levels are also complicated by antibody staining: if samples were permeabilized before staining, intracellular and mis-localized perlecan are included in this metric – and it looks to me as if one feature of neoplastic discs may be the presence of apical or intracellular perlecan.)

RESPONSE: We agree with the reviewer that the z-sections of the Perlecan labeling of undead discs are not always convincing. The reason is naturally occurring folds which often give the impression that there are gaps even in the controls. In contrast, the xy-views of the Perlecan labeling of undead discs always showed gaps and irregularities which are very obvious to score under the microscope. This was also very apparent by the Laminin labeling and the SEM and TEM images. For these reasons, we have replaced the z-sections with the SEM and TEM images in Fig. 1 and the Laminin labelings in Suppl. Fig. S2. To provide frequency of damage, we examined all discs and found that 100% of the undead discs, but none of the control discs, have BM damage at the disc proper side as visualized by Perlecan and Laminin labeling as well as SEM and TEM (summarized in Fig. 1k). We kept the quantification of the Perlecan fluorescence as we believe that these data are important. We also obtained similar data with the Laminin labeling (Fig. S2d).

3. Several other findings lack quantification. To make the observation that hemocytes are activated in some cases but not others, there must be quantification of the activated phenotype (Fig. S3). Same for ROS production in Fig. 2.

RESPONSE: We have included the requested quantifications of the activated hemocytes (Fig. 5i, S1e, S6d, S8g, S13c) and ROS production in Fig. 2 (shown in Suppl. Fig. S6b).

4. It is striking that when Duox is knocked down or catalase is overexpressed (Fig. 3A-C) the disc morphology is so much improved. Fig 3A looks like it is overly compressed, which leads me to speculate about an indirect model for how ROS lead to Mmp2 production: High levels of H₂O₂ would be expected to increase local peroxidase-mediated basement membrane (col4) crosslinking (Bhave et al, 2012; McCall et al, 2014), which would increase basement membrane stiffness (Bhave et al 2017). Stiffness would also be increased by the reduced levels of perlecan (Pastor-Pareja et al, 2011), likely because perlecan is no longer targeted basally (stemming from problems with the apical-basal polarity of the undead cells). In the undead disc, cells are over-proliferating, but constraint by an overly stiff basement membrane could create a JNK-mediated stress signal, triggering the expression of basement membrane cleaving enzymes like Mmp2 to reduce stiffness, a homeostatic circuit that would normally restore the balance between cells and basement membrane. In this case, Mmp2-cleavage of

the basement membrane releases stiffness and gives cells room to grow within the basement membrane; knockdown of Mmp2 inhibits growth. High levels of Mmp2 then recruit hemocytes, which the authors acknowledge has been shown previously.

Of course this model may be incorrect – especially if the basement membrane is not damaged and only perlecan is affected. However, it explains many of the authors observations, and makes stiffness of basement membrane a central question. Matrix stiffness is a central feature of human tumors so it may be worth exploring.

RESPONSE: We do not think that the undead discs are overly compressed. Fig. 3A was just a bad example which we have replaced with a better image.

The stiffness model proposed by the reviewer is very interesting. To explore this possibility, we tested 4 mutant alleles of peroxidasin (pxn) for modification of the adult *ey>hid,p35* undead overgrowth phenotype. However, the adult overgrowth phenotype of the undead flies was not significantly suppressed or enhanced by these alleles. These experiments come with the caveat that the pxn alleles are homozygous lethal, so we were only able to examine this question in a heterozygous pxn mutant background. For that reason, we cannot exclude the possibility that matrix stiffness plays a role in the undead phenotype, and we consider stiffness as something worth considering for future work. On the other hand, because *ey>hid,p35* provides a sensitized condition, many genes – when heterozygous mutant – can modify the *ey>hid,p35*-induced overgrowth phenotype.

5. Although the model put forth by the authors is conceptually clear, many steps of the mechanism are still unsupported, and without at least some of these mechanistic insights, the paper is less impactful. Examples are, how does ROS upregulate Mmp2? Can the authors supply evidence (eg Western blot, zymogram) that the cysteine switch of Mmp2 is activated by ROS? How does JNK upregulate Mmp2 in the undead context but not others? How do hemocytes become activated by ROS? How does Perlecan cleavage/basement membrane cleavage attract hemocytes?

RESPONSE: The reviewer is raising great questions which we also would love to know the answers to. However, while we do speculate about some of these questions in the discussion, we have performed several enzymatic assays to address the second question, is Mmp2 activated by ROS. First, we observed that Mmp2 activity is indeed increased in undead discs (Fig. 4i, Suppl. Fig. S10). Second, the increase of the enzymatic activity of Mmp2 is dependent on ROS and JNK (Fig. 4k, Fig. S10). However, examining if the cysteine switch is activated by ROS is technically very difficult as mutation of this Cys residue to any amino acid will cause enzymatic activity independently of ROS (see reference 75).

We think that the other questions raised by the reviewer are beyond the scope of this paper and might represent important follow-up questions.

6. The distinction between the neoplastic and hyperplastic tumor models would be stronger if it the authors mentioned the criteria underlying this distinction.

RESPONSE: We mention several criteria for neoplastic tumors in line 148 and hyperplastic models in line 158-9.

7. A couple of reference issues: Reference 46 (Llano et al, 2000) has been retracted so probalby

shouldn't be cited. It was recently established that both Mmp1 and Mmp2 have membrane tethered and secreted forms (LaFever et al, 2017), so lines 103-4 can be altered to incorporate this new information.

RESPONSE: We removed reference 46 and updated lines 103-4 with the new reference.

--

Reviewer #2 (Remarks to the Author):

Summary and assessment of novelty and interest:

The work by Diwanji and Bergmann studies the recruitment of macrophages to the undead model for apoptosis induced proliferation in larval imaginal discs.

They claim to have identified that

- 1) ROS and BM damage occur in neoplastic models.
- 2) ROS is required but not sufficient for BM damage
- 3) ROS and JNK are needed to upregulate MMP2 in the undead model
- 4) MMP2 damages the BM leading to macrophage recruitment

Some of the findings in their details are new, (although Suppl Fig S5 D-F were shown already in the Bergmann lab's 2016 Curr Bio paper in Fig 2G-I (Fogarty et al, PMID 26898463)) but the overarching conceptual framework is not very novel. And Fig 2B was already shown in Fig 1E of Perez et al 2017, PMID 28853394

ROS induction in the undead model is known (Fig 1 A-K Fogarty et al). That BM damage occurs has already been shown in a neoplastic model (Fig 4 Pagliarini and Xu, 2003 PMID 14551319, Fig 1 Pastor-Pareja et al DMM 2008, PMID 19048077). The role for ROS in recruiting/ activating hemocytes in the undead model has been shown (Fig 2, Fogarty et al). ROS is already known to induce a different MMP, MMP1 (Fig 3, Fogarty et al). That JNK can induce this same other MMP, MMP1, is known (Fig 2.C,H Uhlrova et al 2006 PMID 170827730). That MMP2 can induce BM damage has been shown (Fig 5 Pastor-Pareja et al DMM 2008, PMID 19048077)). That such BM damage by MMP2 correlates with macrophage recruitment has already been shown in a neoplastic model (Fig 1, Pastor-Pareja et al). That damage to the BM by MMP2 is sufficient to lead to macrophage recruitment to the wing disc or salivary gland has already been shown (Fig 5 Pastor-Pareja et al).

RESPONSE: The reviewer is right that some of the data in our paper were shown previously by us and others (and we have usually cited these papers). However, that was largely done for the sake of completeness or as a starting point for the new findings presented in this manuscript. For example, while our work primarily focused on the role of Mmp2, we thought some readers may also be interested to know about Mmp1, even though that might have been shown somewhere else (assuming that not every reader is so familiar with our work as is the reviewer). Furthermore, the novelty of Figure 2 and associated Figures S5 and S6 lies in the correlation of BM damage, ROS, JNK and hemocyte activation in

neoplastic models, but not hyperplastic models, and not in the content of the individual panels, some of which indeed have been shown by us and others previously (at least for the *Ras^{V12}scrib* model). We believe that these correlations indeed help refining current tumorigenesis models. Nevertheless, in response to the reviewer's comments, we have removed some of the data completely (the Mmp1 data in previous Suppl. Fig. S5B-G) or replaced them with data from a different genotype (the *Ras^{V12}scrib* data in Figures 2, S5 and S6 were replaced by *Ras^{V12}Igl* data).

This submitted paper adds new findings showing the necessity of MMP2 for hemocyte recruitment in this undead model, and data showing that its induction depends on both ROS and JNK signaling. This, I would predict, will be interesting to the field and expand their understanding but will not change current models greatly.

RESPONSE: We would like to point out that our manuscript contains a lot more novel findings than what the reviewer has listed.

1. The BM damage in the undead model.
2. The BM damage depends on ROS and JNK.
3. The asymmetric distribution of Mmp2 in normal eye imaginal discs and the strong induction and basal accumulation of Mmp2 in undead eye discs.
4. The requirement of Mmp2 not only for hemocyte recruitment, but also overgrowth of undead tissue.
5. Hemocyte recruitment is independent of ROS and apoptosis.
6. BM damage correlates with ROS, JNK and hemocyte activation in neoplastic tumor models, but not in hyperplastic growth models.
7. Furthermore, in addition to repeating Pastor-Pareja's finding that Mmp2-inflicted BM damage attracts hemocytes, we now also demonstrate that a good fraction of these hemocytes displays the activated morphology (Figure S13c,d).

Suggested experiments to extend their finding:

They could strengthen their main new conclusion by testing if either ROS treatment or JNK signaling on its own can induce MMP2 expression, and if not, showing the sufficiency of H₂O₂ treatment plus JNK activation to induce MMP2 and hemocyte recruitment. They show in this paper that neither of these treatments on their own can induce basement membrane damage (Fig 3E-F, Fig S4B), however since their data shows that where MMP2 is induced is not always where BM damage occurs (see discussion lines 354-369), they should examine MMP2 expression itself. This is not much work as they already have the line (*salEPV>hepCA, GFP*), and just need to dissect discs and dunk some of them also into H₂O₂.

RESPONSE: We have performed the requested Mmp2 labeling experiment. The H₂O₂ treatment of *sal^{EPV}>hepCA,GFP* wing discs does not induce an increase of Mmp2 expression, shown in Figure R1 below, suggesting that under these in vitro conditions ROS and JNK activation combined are not sufficient for Mmp2 expression. We were unable to check for hemocyte recruitment, because under these ex vivo conditions, the imaginal discs are removed from a hemocyte-containing environment before H₂O₂ exposure.

Figure R1: ROS and JNK together are not sufficient to induce Mmp2 expression

(a-d') Representative examples of wing imaginal discs with wild-type (a-b') or hyper-activated JNK (*hep^{CA}*) (c-d') incubated *ex vivo* in Schneider's media with 0 μM H₂O₂ (a, c) or 50 μM H₂O₂ (b, d) and labelled with antibody against Mmp2. Scale bars, 100 μm

(e) Quantification of Mmp2 intensity in (a-d) taken from maximum intensity projections. Signal intensity per μm² ± SEM analyzed by two-way ANOVA with Tukey's multiple comparisons test. n.s. = no statistical significance. (n=4 (*saI^{EPV}*>*GFP* 0 μM H₂O₂), 9 (*saI^{EPV}*>*GFP* 50 μM H₂O₂), 4 (*saI^{EPV}*>*hep^{CA}*, *GFP* 0 μM H₂O₂) and 15 (*saI^{EPV}*>*hep^{CA}*, *GFP* 50 μM H₂O₂) discs of each genotype analyzed)

Major problems with data:

I am not convinced by the data presented in some places due to a lack of quantitation and/or a discordance between the quantitation presented and the supporting immunofluorescent panels.

Perlecan data:

Most strikingly, the data provided analysing basement membrane damage or lack thereof do not clearly match the conclusions drawn or the quantitation made (when quantitation is provided, which is not always the case). In Figure 1 for example, it is hard to reconcile the staining shown for Perlecan in B'-D' or B''-D'' with the quantitation shown in E. They just don't seem to match at all; the changes in B''-D'', particularly between C'' and D'' are very subtle and not clearly evident, not matching the quantitation shown in E. Looking at B-D' one would conclude that there is a reduction in Perlecan levels already in the ey>p35 control, again not matching what is in the quantitation. I have similar issues for Figs 3A-C'' where the changes in the DP shown in the quantification are not obvious to me and where the Perlecan staining in the PE in panel C'' looks completely absent compared to A'' while being at the same level in the quantification. Same issues for Fig 5 A-C''.

This lack of matching between quantitation and images for Perlecan is particularly striking when one compares it to the data for Mmp2 shown in Figure 4, where the again two fold differences in levels in the DP side shown in the quantitation in Fig 4D are completely obvious and correlate exactly with what

is shown in panels 4A-C and A''-C''.

I tend to believe the data in quantitation of multiple samples over a single example, however if there is no single set of examples that roughly matches the average of the quantitation, that is strange.

RESPONSE: The point we want to make in Figure 1 is that all experimental (undead) discs have BM damage based on Perlecan labeling, while none of the control discs do. The quantification of the Perlecan staining in 1E was our attempt to measure this; however, as reviewer #1 also pointed out, these measurements may not be adequate to document this. We agree with the reviewer that the z-sections in Fig. 1B''-D'', 3A''-C'' and 5A''-C'' do not always show the damage in an obvious manner, but we respectfully disagree with the reviewers' assessment that the Perlecan labeling in Figure 1B'-D'' and the quantification in E don't match at all. The reason why the reviewer may think that, is that there are naturally occurring folds which often give the impression that there are gaps in the z-sections even in the controls. The same applies to Fig. 3A-C'' and 5A-C''. However, the xy-views of the Perlecan labeling of undead discs always show gaps and irregularities which are very obvious under the microscope. This is also very obvious by the new Laminin labeling and the SEM and TEM images (see new Fig. 1, Suppl. Fig. S2 and S3). For these reasons, we have replaced the z-sections in Figure 1 with the new SEM and TEM images and show the new Laminin labeling in Suppl. Fig. S2. We also have added the number of discs examined for each genotype (Fig. 1k) demonstrating that all experimental discs display irregularities in Perlecan, Laminin, SEM and TEM labelings, while none of the control discs do.

There is also no comprehensive description of how the Perlecan (or MMP2) quantitation was conducted, to make me sure that difference between the data shown and quantitation isn't due to some problem with the quantitation. Was the quantitation done in the region of overgrowth? Were equivalently sized regions were outlined from a single z stack focusing on the basal side from the different samples and had their level of fluorescence assessed? Was there some internal control to normalize for differences in general staining between samples by assessing the level of Perlecan in a region where no change would be expected (ie a part of the disc that isn't undead). If that isn't appropriate because the changes appear to extend beyond the region of the hid p35 expression can one use the staining for some other ubiquitous protein as a control for differences in fixation that might alter antibody staining generally?

RESPONSE: We have provided a more detailed description of how these quantification were done in the Methods section.

Request 1, additional production of extant data and textual change:

Please provide more of the data that was quantitated to clarify this issue for me, and/or provide me with a convincing explanation for what is going on. Please improve the description of Perlecan quantitation in the methods.

Let's assume that the quantitation is fine and there is some reason why the pictures shown don't match it very obviously. Then, given that the pictures are not easy to interpret, having Figures that provide only one picture of Perlecan without quantitation is also problematic.

RESPONSE: As pointed out above, the point of the Perlecan labeling was to demonstrate that BM damage only occurs in the undead discs. These pictures are very easy to interpret and we have now more supportive evidence from the Laminin labelings and the SEM and TEM images (see summary of data in Fig. 1k).

Request 2, additional analysis and textual change, Fig 3, S2 and S4:

Please quantitate the level of Perlecan in the H₂O₂ treated case (corresponding to what is shown in Fig 3E, F), to be sure there is no change.

Please quantitate Perlecan levels for Figure S2 A",B', where the pictures shown do not at all support the conclusion stated.

Please quantitate the level of Perlecan for Fig S4 A-C to anchor the conclusion that JNK is required for BM damage and include the control for the salEPV panel S4B.

For all this quantitation for these panels list how many discs were examined and do the statistics.

RESPONSE: All the requested quantifications were done and shown in Figs 3I, S4c, S7h, respectively, including the statistics and n numbers in the figure legends.

Request 2: Further analysis, Fig 2

I didn't find Figure 2 on the correlation of BM and ROS very convincing, and found this figure mostly useful for assessing if overgrowth on its own can cause BM damage (which I think would make it a better supplemental Figure). But in any case they need to quantitate whatever they show (BM damage and ROS if both are still included).

RESPONSE: All requested quantifications are now shown in Fig. S6.

Request 3, Further analysis and textual change related to MMP2 data in Fig 4:

a) It appears as if there is some upregulation of MMP2 in Fig 4B (in the *ey>p35* control) based on the whole disc view, even though this is not seen in the close up in B'. Please clarify. This might fit given that it is known that there is already hemocyte recruitment in the *ey>p35* case (see below, Hemocyte point c).

RESPONSE: There is indeed an increase of *Mmp2* expression in the *ey>p35* control. This is quantified in Suppl. Fig. S9. However, despite this increase, there is no accumulation of *Mmp2* protein at the basal side of the DP of *ey>p35* discs, and that is what is quantified in Fig. 4D. We are pointing this out in the Results section.

a) In Fig 4C they see induction of MMP2 throughout the entire disc, not just the portion that is undead. This does not fit their model as I understand it in which upregulation of JNK signaling in the undead cell (Fig 7) leads to higher levels of MMP2 in those undead cells. They explain in the discussion how localized ROS could potentiate MMP2 activity to allow the localized presence of BM damage despite the uniform MMP2 induction. However they don't explain how localized JNK and localized ROS induction lead to ubiquitous MMP2 expression. There could be some other component downstream of both ROS and JNK that diffuses broadly to induce MMP2 expression in the whole disc (unless something is wrong with their staining). To make sure the MMP2 Ab isn't binding nonspecifically in undead discs, they should do a control staining in the background in which the undead disc also expresses RNAi against MMP2 (the line they already have, given data in Fig 5C). If the antibody is truly binding MMP2 then they should see downregulation of MMP2 Ab staining in the undead region of the disc. If this control is fine and if they agree with my interpretation of their results as described above, modify Fig 7 accordingly.

RESPONSE: We were also very surprised to see the uniform upregulation of Mmp2 across the entire disc as shown in Fig. 4C. However, we have confirmed this result multiple times (Suppl. Fig. S9c shows another example of the uniform upregulation of Mmp2 across the entire disc, so this result is real). The requested experiment (*ey>hid,p35;Mmp2* RNAi) was already shown in previous Suppl. Fig. S6F which is now Suppl. Fig. S9d (quantified in S9e). It shows strong reduction of Mmp2 protein in the entire disc. The easiest interpretation of this result is that Mmp2 protein can diffuse into other parts of the disc where it is not expressed. However, as suggested by the reviewer, we do not exclude the possibility that there are additional factors which mediate the expression of Mmp2 downstream of JNK.

b) In Fig 4 D they only quantify MMP2 in the DP. It appears as if MMP2 is upregulated in DP and PE based on the picture in 4C. To assess this they should also quantify expression in the PE.

RESPONSE: We have completed the quantification, shown in Suppl. Fig. S9b.

Request 4, further analysis hemocyte data:

a) Fig 5E-G Please quantitate the amount of hemocyte attachment. It appears as if there is a reduction in the Mmp1 RNAi case as well based on the panel shown.

RESPONSE: We quantified the amount of hemocytes attached to the discs of these genotypes (see Fig. 5h). There is no significant decrease of hemocyte attachment in *ey>hid,p35; Mmp1* RNAi discs.

b) There is no clear data to substantiate the claim that hemocytes are inactive in the hyperplastic model shown in S3 or in Fig S4C when there are fewer hemocytes present. Please show a close up to show lack or presence of extensions and quantitate their extent.

RESPONSE: Close-ups of hemocytes with activated morphology are shown in Fig. 5e'-g', S1c'', S8d',e' and S13c. We also quantified these data, shown in Fig. 5i, S1e, S6d, S8g, S13c.

c) in this paper there is no examination and quantitation of the *ey>p35* control for hemocyte recruitment, only *ey-GAL4* compared to *ey>p35 >Hid* (Fig S1). The Fogarty et al paper Fig 3A showed that hemocytes are already recruited in *ey>p35* but that in *ey>p35>hid* hemocytes shift to an activated state.

This raises the question as to whether MMP2 is required also for recruitment of hemocytes in the *ey>p35* control. They should analyse this. If yes, then this MMP2 pathway might not be particularly specific for recruitment in the undead model but more generally induced in response to blocking apoptosis. This might also fit with the data shown in Fig 1C, C'' where it appears as if there is a downregulation of Perlecan in this control. If this is the case they would need to rewrite the paper to focus on macrophage recruitment to cells in which apoptosis is blocked and extend the dissection of MMP2 induction in the *ey>p35* system. They need to shift the framing, which is currently about understanding recruitment to a tumor model.

RESPONSE: This is a legitimate concern of the reviewer. However, *Mmp2* RNAi does not reduce the recruitment of hemocytes to *ey>p35* discs, so the concern of the reviewer is unwarranted. This is shown

here as Figure R2 only to the reviewers as it is not adding anything important to the main point of the manuscript:

Figure R2: *Mmp2* is not required for hemocyte recruitment to *ey>p35* discs

(a-b'') Hemocytes (labeled by NimC; red in a, b; grey in a', a'', b', b'') in *ey>p35* (a) and *ey>p35; Mmp2* RNAi discs (b). Similar to *ey>p35* discs, discs with knockdown of *Mmp2* still have hemocytes present in tight clusters (magnified in a'', b'') at the boundary of the differentiating photoreceptor neurons (labeled by ELAV; green in a, b). Scale bars, 100 μ m (a-b') and 50 μ m (a'', b'')

(c) Quantification of hemocyte recruitment in *ey>p35* and *ey>p35; Mmp2* RNAi discs. Total number of hemocytes were analyzed by Unpaired T test. n.s. = no statistical significance (n=20 (*ey>p35*) and 14 (*ey>p35; Mmp2* RNAi) discs per genotype analyzed).

Request 5: Further analysis, Fig 6

Please quantitate the DHE levels and the DCp-1 staining for all genotypes.

List n and do statistics.

RESPONSE: This was done (Figs S6b, S12d, S13e, S13f).

Textual changes:

1) In the introduction in line 53, they don't mention that MMP2s have been shown to be sufficient to recruit hemocytes (shown in Pastor-Pareja et al, their ref 12, which is only mentioned glancingly as showing hemocyte recruitment to tumors.)

RESPONSE: We have now cited this paper by mentioning that BM damage is sufficient for hemocyte recruitment. However, we have done this in line 93-4 of the Introduction, because it fits better there with the flow of information.

2) line 55, I wouldn't say "mimicking", I would say "analogous to"

RESPONSE: We made the suggested change (now in line 54).

3) In introduction lines 70-73 they reference previous findings on ROS and JNK driving AiP but don't mention the previous data showing that ROS is required for hemocyte recruitment in Fig 3E, F Fogarty et

a). They should.

RESPONSE: This has been corrected.

4) In Fig 2, there is little evidence that ROS and BM alterations overlap, either with the region of the tumor (in the RasV12 case) or with each other (See Fig 2B where there are strong DHE clusters both far away from or neighboring tumor clones as well as some within clones.) Thus when saying (line 167) that DH puncta are observed they should add “in and around tumor clones”. I would also qualify their statements about correlation (line 171) to say, “the appearance of BM damage and ROS in the disc seems to occur coincidentally upon the induction of neoplastic growth. Since they have not assessed both ROS and BM damage in their AiP model, they should strike lines 171-173.

RESPONSE: We made the suggested additions in line 167 (now line 174) and line 171 (now 177-8). We did not strike lines 171-173, because we have assessed both ROS and BM damage in this AiP model.

5) line 220, should be S4B,B” not S4D,D”

RESPONSE: Thank you for pointing this out. We corrected the mistake.

6) line 233, and 973. They can’t conclude in Fig S4 that hemocytes are not necessary for damaging the BM if hemocytes are still present, which they demonstrate. Please soften the conclusion to saying damage to the BM is not linearly correlated with the amount of hemocytes (if the quantification shows equivalent levels of BM in the ey>hid and the ey>hid srpneo45/+ situation).

RESPONSE: As requested, we softened our language and added that “there is no linear relationship between BM damage and the amount of hemocytes attached to the discs”.

6) Text line 846-7, I think they mean “discs still had gaps as in the mock”

RESPONSE: Yes, that is what we meant. Thank you for pointing this out.

7) They explain why they shift to the wing disc model for Fig 6 in the text, but they do not explain why they used this alternate model for the data shown in Fig S2 and Fig S4. Please do so.

RESPONSE: We provided the explanation in the context of Fig S2 and S4 (now S4 and S7).

8) They should list in the methods the exact genotype of the stocks used in all Figure panels

RESPONSE: There are four major genotypes in this manuscript: ey-Gal4, ey>p35, ey>hid,p35 and sal^{EPV}>GFP. The complete genotype of these stocks is presented in the methods. All other genotypes contain an additional UAS transgene which is self-explanatorily indicated in the panels of the figures. The only exception are the MARCM crosses in Fig. 2, S5 and S6. The detailed genotypes of these individuals are listed in the legend to Figure 2.

--

Reviewer #3 (Remarks to the Author):

This is a fascinating study from Diwanji and Bergmann that investigates the mechanism by which *Drosophila* hemocytes are recruited to undead cells to trigger a signalling loop that ultimately leads to tissue overgrowth via AiP. Using the powerful genetics available in the fly the study uncovers a mechanism whereby ROS production together with JNK signalling within the undead cells leads to the activation and accumulation of MMP2 which in turn leads to a loss in basement membrane integrity. They also show evidence that it is this basement membrane damage that is responsible for the recruitment of hemocytes to the undead cells which play a key role in the signalling loop. The signalling they uncover using the undead – AiP model appears to hold true in neoplastic tumours. This paper is well written, the experiments are well designed and the data of good quality. The hemocyte recruitment and ‘activated morphology’ need to be quantified throughout the paper but if the authors can address that together with the following points then I think it would be an excellent paper for Nature Communications.

RESPONSE: We have quantified the recruitment and activation of hemocytes throughout the paper, shown in Fig. 5e’-g’,i, S1c’’,e, S6d, S8d’,e’,g and S13c.

1. The authors refer to hemocytes at tumours and undead cells as having undergone ‘alternative activation’ which provides them with a TAM-like activity. I think that is a stretch too far. There is little evidence of these cells really behaving as TAMs and their ‘activated morphology’ is at present not convincing and needs more rigorous quantification (see below).

RESPONSE: We softened our language regarding the comparisons of hemocytes with TAMs. We have removed the term ‘alternative’ with respect to hemocyte activation and also avoided to compare TAMs directly with activated hemocytes.

2. Supp Figure 1 A’’ apparently shows a high magnification zoom of A’ but the sample looks completely different. The same is true of B’ and B’’. The authors need to highlight the area being magnified on the lower mag image. For both these images as well as Figure 5F the ‘activated morphology’ of hemocytes is not clear. The authors should show higher magnification images to clearly demonstrate the protrusions on activated hemocytes and quantify this protrusive morphology as well as hemocyte recruitment (hemocyte number) throughout the paper. The current images are not very convincing/clear

RESPONSE: We changed the images in Figure S1, show high-mag images of activated hemocytes (Fig. 5e’-g’, S1c’’, S8d’,e’ and S13c), and performed the quantification of the activated hemocytes throughout the manuscript (Fig. 5i, S1e, S6d, S8g, S13c).

3. The authors show that providing either ROS or JNK signalling alone is insufficient to cause basement membrane damage and therefore conclude that both are required. They should directly show this using their ex vivo assay by exposing discs that express hepCA to hydrogen peroxide and quantifying basement membrane disruption

RESPONSE: Thank you for proposing this experiment. Exposure of *sal*^{EPV}>*hep*^{CA} to H₂O₂ does indeed cause an irregular appearance of the BM. However, the quantification of the Perlecan fluorescence does not indicate a significant loss of Perlecan fluorescence suggesting that under these *ex vivo* conditions the effect of H₂O₂ on *hep*^{CA}-expressing discs on the BM is weaker than in undead eye discs. These data are shown in Fig. 3h-l.

4. The authors use UAS-MMP2 overexpression to generate basement membrane damage in the absence of ROS and still get hemocyte recruitment. What happens if they generate ROS from undead cells but block basement membrane damage? They have in fact already done this in the experiment described in Figure S4 (i.e JNK-DN *ey>hid p35*). Are hemocytes recruited to the undead cells in this experiment?

RESPONSE: Under *JNK*^{DN} *ey>hid,p35* conditions, there are no or only very few hemocytes present at the discs. This is shown in Figure R3 below (NimC labels hemocytes).

Figure R3: Hemocytes are not recruited to *ey>hid,p35; JNK*^{DN} discs

Very few hemocytes (labeled by NimC; red (a), grey (a')) are present on *ey>hid,p35* discs upon inactivation of JNK. Basement membrane (labeled by Perlecan; green (a)) is not damaged under these conditions. Scale bars, 100 μm.

5. The title doesn't seem to fit the take home message. The basement membrane damage requires both ROS and JNK signalling so the title should either include JNK as well as ROS or they should lose ROS to simply state 'MMP2-mediated basement membrane damage recruits macrophages to overgrown tissue in drosophila'

RESPONSE: We changed the title of the manuscript to better reflect the take-home message.

Minor points

1. Why do the authors use the wing disc for the experiment in Figure S2 rather than the eye disc as is used for the rest of the paper?

RESPONSE: The reason for using the wing disc in the experiments of Figure S2 (now S4 and also in Figs 3, 6 and S7) is that expression of *hid*, *hep*^{CA} (and *Mmp2*) using the *ey-Gal4* driver causes early larval lethality which prevents us from doing this experiment. Because the wing-specific *sal*^{EPV}-*Gal4* driver comes on later in development (end of 2nd larval instar), larvae expressing *hep*^{CA} survive to the end of 3rd larval instar when we usually examine these discs. This is now also better explained in the manuscript.

2. line 378. shouldn't 'cryptic receptors' read 'cryptic ligands'?

RESPONSE: It could be either way. What we mean is that in response to BM damage, there are some membrane-bound proteins exposed on the surface of the undead cells which are now accessible by hemocytes. We referred to the membrane-bound proteins as 'cryptic receptors' which are bound by ligands from hemocytes, but one could also make the opposite case. We did not change the text.

3. The authors should cite the following paper in their discussion that demonstrated that upon induction of apoptosis, lymphoma cells express and process MMP2 showing clear parallels between vertebrate studies and the data the authors present here in the fly: Ford et al, 2015, Current Biology 25(5):577-88

RESPONSE: This is indeed an important paper which we are now discussing and citing (ref. 5).

Reviewers' Comments:

Reviewer #1:

Remarks to the Author:

The manuscript by Dewanji and Bergmann is much improved with the addition of new ways of measuring basement membrane damage and the quantification of data throughout the study. The data in the paper show—

1. In *Drosophila* larvae, undead discs and neoplastic tumors have basement membrane damage. This is shown convincingly for the undead discs.
2. Undead discs and neoplastic tumors have a cluster of related attributes: basement membrane damage, upregulation of ROS, upregulation of JNK, and accumulation of hemocytes many of which are activated.
3. Both ROS and JNK are required in the discs for BM membrane damage. In contrast, the extent of hemocyte accumulation is not correlated with BM damage.
4. Although neither ROS nor JNK alone is sufficient to cause BM damage, the combination of the two is sufficient to damage discs *ex vivo*. (this is new)
5. Mmp2 protein levels and activity are increased non-autonomously in undead discs, dependent on both JNK and ROS in the discs. Mmp2 is required for BM damage. However, JNK and ROS are not sufficient to increase Mmp2 protein levels (from a figure for reviewers)
6. As expected from previous studies, the overexpression of Mmp2 is sufficient to recruit hemocytes (although this figure, Fig. 6, is missing from the manuscript).

The discussion proposes a model mechanism for how BM damage is induced and hemocytes are recruited, yet this model is surprisingly disconnected from these data. In the model, hemocytes have an early critical role, being activated by ROS to secrete Eiger (for which there is no data). Yet the role of hemocytes is not supported by points 3 and 4 above. I guess the problem is how the JNK pathway gets upregulated in the undead discs, but perhaps that is a feature of "undeadness" itself. If so then these levels of ROS and JNK could activate Mmp2, which could then propel the feed-forward loop.

There are still technical problems with the manuscript, some of which can be fixed by text editing and some which require new analysis.

1. One persistent problem from the last version of the manuscript is the use of perlecan staining levels alone to assess BM damage for many experiments. I had requested alternative methods; reviewer 2 had requested internal controls and clarification on their methods, including the region measured. For many figures, these requirements are still not met. This is especially problematic on Fig. 3, showing BM damage from perlecan staining when JNK and ROS are experimentally upregulated. The quantification of the authors is not compelling because the Methods indicate that perlecan staining was measured throughout the disc, whereas only the small *Sal>gal4* regions should be measured. This seems like an exciting result, downplayed by the authors. Definitive assays – like SEM -- would help enormously here.
2. In Fig. 4, the panels I,J,K appear to show three replicates of an MMP activity assay. But looking carefully at the curves, it appears to be one experiment reproduced three times to make different comparisons. Show this data only once if it is one replicate. Does Nature Communications allow publication of experiments where $n=1$?
3. Fig. 6 is missing -- by some unfortunate accident, Fig. 5 was duplicated in the place of Fig. 6.
4. Fig. S4 is not convincing. A positive control of an undead disc is needed, imaged in the same way, to show what BM damage looks like in this assay. The quantification is not believable because the entire disc was measured, not the *Sal* region.
5. For S6, there are no controls for panels b or e. (Controls for panels c and d may be in Fig. S1, but are they comparable? If so, point that out).

Reviewer #2:

Remarks to the Author:

In this revision the authors have addressed many of the issues raised but have not addressed other essential ones at all, specifically for Figures 3 and 5 that are meant to support their claims that JNK and ROS and MMP2 are required for BM damage. They also have made misleading statements about some of their new data which does not, as they claim, support the idea that JNK and ROS are sufficient to induce BM damage. I do not regard this as a sufficient revision to prove all of their initial points. I do not believe this paper is rigorous enough nor provides enough of an advance to warrant publication in Nature Communications.

See detailed comments below:

Figure 1 much better now. SEM data very convincing for the DP and PE. The authors have provided much easier to interpret pictures of the whole disk and very convincing laminin staining. My concerns for this Figure have been beautifully addressed.

Fig 3A'-C', A''-C'' they haven't validated their findings with the SEM or with laminin staining, nor have put more example pictures for the reviewers to show how their quantification in e could be justified. They have just removed the panels I that I found didn't match the quantification. The same comment holds for Fig 5 A-C-, A''-C''

The objections that I made in the original review for these figures thus have not been addressed.

The description of the quantitation has not been made more complete. The points I raised regarding an internal control, whether equivalently sized regions are being analysed, or if the analysis occurs within the hid p35 expression area were not addressed.

They have provided the quantification I requested for previous Fig 3, S2 and S4 and Fig 2 and Fig 6.

Textual request 4)

They did not make the textual changes in previous line 171 to decrease the statements about the correlation of the BM damage and the ROS given that they have not demonstrated this, even though they stated that they did make these changes.

The authors state that they now show in the revised Figure 3 that JNK and ROS together are sufficient to induce BM damage (lines 232-4). This is not supported by the data they present. While they claim that this is shown in Figure 3 j versus k this is not obvious in these pictures. The quantitation shown in Figure 3l shows no significant difference in the levels of perlecan between control discs expressing a JNK pathway activating constructs and those treated with H₂O₂. Their statement in lines 236-7 about the quantitation is disingenuous: "The damage under these conditions is not as strong as in undead discs", when in reality their quantitation shows no evidence for damage at all. Thus the authors must alter their statement in line 232-4 and 236-7 to state that the in vitro situation does not recapitulate what they observed in vivo or remove this data entirely.

They must also change the title of Figure 3 in Line 900 in the Figure legends to state that ROS and JNK are required for BM damage of imaginal discs, removing the phrase "and together cause"

Seeing no BM damage under these conditions would fit with the data they show in Figure R1 indicating that in vitro JNK and ROS are not sufficient to induce MMP2, given that the authors show MMP2 is required for BM damage elsewhere in the paper.

Their proof is not conclusive so they should change the Fig S8 title (line 127 in supplementary Figures) to "Hemocyte numbers do not correlate with damage of undead discs"

They gave us the wrong Fig 6. It is the same as Fig 5, but I don't think there were any alterations there in the revision.

Reviewer #3:

Remarks to the Author:

The authors have addressed all my concerns and the paper should now be accepted for publication.

We would like to thank the reviewers who took valuable time off their schedules to review the manuscript and for their insightful comments and suggestions to further improve it. We hope that we now have addressed all concerns of the reviewers to their satisfaction. We have made several textual changes, especially to clarify the quantifications in the Methods section and have added more data wherever requested, including a new figure here just for the reviewers. Please see the detailed point-by-point response below.

Reviewers' comments:

Reviewer #1 (Remarks to the Author):

The manuscript by Dewanji and Bergmann is much improved with the addition of new ways of measuring basement membrane damage and the quantification of data throughout the study. The data in the paper show—

1. In *Drosophila* larvae, undead discs and neoplastic tumors have basement membrane damage. This is shown convincingly for the undead discs.
2. Undead discs and neoplastic tumors have a cluster of related attributes: basement membrane damage, upregulation of ROS, upregulation of JNK, and accumulation of hemocytes many of which are activated.
3. Both ROS and JNK are required in the discs for BM membrane damage. In contrast, the extent of hemocyte accumulation is not correlated with BM damage.
4. Although neither ROS nor JNK alone is sufficient to cause BM damage, the combination of the two is sufficient to damage discs *ex vivo*. (this is new)
5. Mmp2 protein levels and activity are increased non-autonomously in undead discs, dependent on both JNK and ROS in the discs. Mmp2 is required for BM damage. However, JNK and ROS are not sufficient to increase Mmp2 protein levels (from a figure for reviewers)
6. As expected from previous studies, the overexpression of Mmp2 is sufficient to recruit hemocytes (although this figure, Fig. 6, is missing from the manuscript).

The discussion proposes a model mechanism for how BM damage is induced and hemocytes are recruited, yet this model is surprisingly disconnected from these data. In the model, hemocytes have an early critical role, being activated by ROS to secrete Eiger (for which there is no data). Yet the role of hemocytes is not supported by points 3 and 4 above. I guess the problem is how the JNK pathway gets upregulated in the undead discs, but perhaps that is a feature of “undeadness” itself. If so then these levels of ROS and JNK could activate Mmp2, which could then propel the feed-forward loop.

RESPONSE: We agree with the reviewer that the problem is how the JNK pathway is initially activated because several important steps such as Mmp2 activation and hemocyte recruitment are dependent on JNK, but appear to be happening upstream of JNK. This is possible because of the amplification loop. However, the problem of the initial JNK activation remains. To make the model clearer, we have added another panel to Fig. 7B (the middle panel). We are proposing that ROS provide the initial signal for activating the naïve hemocytes already present on the disc. In one of our previous papers (Fogarty et al., 2016 (ref. 17)), we showed that Eiger is secreted by hemocytes during AiP. Eiger provides a signal for JNK activation in undead cells. That’s why we put Eiger in the model. However, to avoid confusion, we have now removed Eiger from the Figure and just mention it in the text only. In the same reference, we also showed that hemocytes are necessary for AiP-induced overgrowth. We are mentioning this in the Introduction (line 55/56). We propose that this early induction of JNK activation leads to upregulation of Mmp2 and Hid, thus propelling the amplification loop. However, this is just a model and we are aware that other models might be possible, too. As new data are coming in, we may need to revise our model.

There are still technical problems with the manuscript, some of which can be fixed by text editing and some which require new analysis.

1. One persistent problem from the last version of the manuscript is the use of perlecan staining levels alone to assess BM damage for many experiments. I had requested alternative methods; reviewer 2 had requested internal controls and clarification on their methods, including the region measured. For many figures, these requirements are still not met. This is especially problematic on Fig. 3, showing BM damage from perlecan staining when JNK and ROS are experimentally upregulated. The quantification of the authors is not compelling because the Methods indicate that perlecan staining was measured throughout the disc, whereas only the small *Sal>gal4* regions should be measured. This seems like an exciting result, downplayed by the authors. Definitive assays – like SEM -- would help enormously here.

RESPONSE: For almost all of the experiments, we also showed laminin labelings in the supplement to support the perlecan labelings including quantifications (Suppl. Figs. S2, S3, S7 and S11 in the previous submission; now S2, S3, S7 and S12).

We agree with the reviewer that the quantifications needed more clarification and detailed description of the methods used. In the case of the *sal>Gal4* experiments, all quantifications were done only in the *sal^{EPV}* domain. This is now specifically mentioned in the Methods. We apologize if we caused any confusion by not specifically mentioning that. For the Perlecan and Laminin quantifications, we have used the fluorescence at the PE side as an internal control because here the BM is not damaged (Suppl. Fig. S3j,k)

2. In Fig. 4, the panels I,J,K appear to show three replicates of an MMP activity assay. But looking carefully at the curves, it appears to be one experiment reproduced three times to make different comparisons. Show this data only once if it is one replicate. Does Nature Communications allow publication of experiments where n=1?

RESPONSE: Each of the experiments in panels 4i, j, k were performed three independent times, so n=3. We could have presented the data all in one panel as shown below in Figure R1. However, for the ease of data presentation, we split these data into three different groups based on the topic and thus different panels. We believe this display will help the readers to better follow the flow of information and clearly understand the results. Only the *ey>hid,p35* data (red line) are present in all three panels. Because all these experiments were performed in parallel, we think that is ok to do.

For the reviewer, we have included the composite panel here, and can definitely exchange panels 4i, l, k with this panel if the reviewer so wishes.

Figure R1: Mmp2 activity assay. Mmp2 activity measured by fluorometric assays using fluorescein-linked DQ-gelatin substrate. The different genotypes are color-coded as indicated on the right. *ey>hid,p35* disc lysates (red line) show increased cleavage of DQ-gelatin substrate indicated by a marked increase in fluorescence compared to controls. This increased cleavage activity is lost upon knockdown of *Mmp2* (black) and is comparable to *Mmp2* overexpression (blue). Mmp2 activity in

ey>hid,p35 undead discs is dependent on ROS and JNK as the fluorescence intensity of the substrate decreases upon loss of ROS or downregulation of JNK activity.

3. Fig. 6 is missing -- by some unfortunate accident, Fig. 5 was duplicated in the place of Fig. 6.

RESPONSE: We apologize for this oversight and have now included Fig. 6 in this submission. However, Figure 6 did not change compared to the first submission.

4. Fig. S4 is not convincing. A positive control of an undead disc is needed, imaged in the same way, to show what BM damage looks like in this assay. The quantification is not believable because the entire disc was measured, not the Sal region.

RESPONSE: We have now included the positive undead disc control (*sal^{EPv}>hid,p35*) labelled with Perlecan in Fig S4c. As requested, we quantified these discs only in the *sal^{EPv}* region. In the undead discs, we see presence of holes and gaps, and the Perlecan intensity is much lower than the control discs.

5. For S6, there are no controls for panels b or e. (Controls for panels c and d may be in Fig. S1, but are they comparable? If so, point that out).

RESPONSE: We have now added controls to all the panels in Fig S6.

--

Reviewer #2 (Remarks to the Author):

In this revision the authors have addressed many of the issues raised but have not addressed other essential ones at all, specifically for Figures 3 and 5 that are meant to support their claims that JNK and ROS and MMP2 are required for BM damage. They also have made misleading statements about some of their new data which does not, as they claim, support the idea that JNK and ROS are sufficient to induce BM damage. I do not regard this as a sufficient revision to prove all of their initial points. I do not believe this paper is rigorous enough nor provides enough of an advance to warrant publication in Nature Communications.

See detailed comments below:

Figure 1 much better now. SEM data very convincing for the DP and PE. The authors have provided much easier to interpret pictures of the whole disk and very convincing laminin staining. My concerns for this Figure have been beautifully addressed.

RESPONSE: Thank you.

Fig 3A'-C', A''-C'' they haven't validated their findings with the SEM or with laminin staining, nor have put more example pictures for the reviewers to show how their quantification in e could be justified. They have just removed the panels I that I found didn't match the quantification. The same comment holds for Fig 5 A-C-, A''-C''

The objections that I made in the original review for these figures thus have not been addressed.

RESPONSE: The reviewer must have missed that Laminin stainings as a validation of the Perlecan stainings in Fig. 3a-d were provided in the previous revision in Suppl. Fig. S7a-d, quantified in S7e, and the ones in Fig. 5a-c were validated in Suppl. Fig. S11a-c, quantified in S11d (now S12a-d).

Below, we have now also included some additional example pictures for the different genotypes from Fig. 3a-d and Fig. 5a-c here for the reviewers to see (Figure R2). We hope that these additional examples will address the issues the reviewer has with the data presented in the manuscript.

ey>hid,p35

Figure R2. Shown are three additional examples of each genotype indicated on the left of the panels to support the data in Fig. 3a-d and Fig. 5a-c. These discs were labeled with Perlecan, NimC (except *ey>hid,p35* alone) and ELAV antibodies.

The description of the quantitation has not been made more complete. The points I raised regarding an internal control, whether equivalently sized regions are being analysed, or if the analysis occurs within the hid p35 expression area were not addressed.

Response: For all *ey>hid,p35* discs, the entire discs were analyzed. For all the *Sal^{EPV}>Gal4* experiments, equivalently sized *sal* domains were measured. In our assays, BM is not affected at the PE and thus the Perlecan or Laminin intensity at the PE remains constant. We have therefore used the intensity at PE as an internal control for the intensity at DP for each disc. This is also explained in the Methods section.

They have provided the quantification I requested for previous Fig 3, S2 and S4 and Fig 2 and Fig 6.

Textual request 4)

They did not make the textual changes in previous line 171 to decrease the statements about the correlation of the BM damage and the ROS given that they have not demonstrated this, even though they stated that they did make these changes.

RESPONSE: Contrary to what the reviewer said, we did make that change, almost word for word. In the first review, the reviewer requested: I would also qualify their statements about correlation (line 171) to say, "the appearance of BM damage and ROS in the disc seems to occur coincidentally upon the induction of neoplastic growth". Here is what we wrote in the revised manuscript: "Together, these data suggest that BM damage and ROS production correlate in the disc, and this correlation coincides with the induction of neoplastic growth." (lines 177-178 in the last revision, now lines 183-184).

The authors state that they now show in the revised Figure 3 that JNK and ROS together are sufficient to induce BM damage (lines 232-4). This is not supported by the data they present. While they claim that this is shown in Figure 3 j versus k this is not obvious in these pictures. The quantitation shown in Figure 3l shows no significant difference in the levels of perlecan between control discs expressing a JNK pathway activating constructs and those treated with H2O2. Their statement in lines 236-7 about the quantitation is disingenuous: "The damage under these conditions is not as strong as in undead discs", when in reality their quantitation shows no evidence for damage at all. Thus the authors must alter their statement in line 232-4 and 236-7 to state that the in vitro situation does not recapitulate what they observed in vivo or remove this data entirely.

They must also change the title of Figure 3 in Line 900 in the Figure legends to state that ROS and JNK are required for BM damage of imaginal discs, removing the phrase "and together cause" Seeing no BM damage under these conditions would fit with the data they show in Figure R1 indicating that in vitro JNK and ROS are not sufficient to induce MMP2, given that the authors show MMP2 is required for BM damage elsewhere in the paper.

RESPONSE: While we do see some minor BM damage under conditions of ROS and JNK activation in the *sal^{EPV}* domain (Fig. 3k''), this does not translate in a significant change in Perlecan staining in Fig. 3l. We therefore followed the advice of the reviewer and changed our language in the manuscript. We basically state now that ROS and JNK are required, but not sufficient for BM damage. To provide a rational explanation of this result, we now show the previous Figure R1 in the supplement as Suppl. Fig. S10.

Their proof is not conclusive so they should change the Fig S8 title (line 127 in supplementary Figures) to "Hemocyte numbers do not correlate with damage of undead discs"

RESPONSE: Thank you for pointing this out. We truly forgot to make this change. We made it now.

They gave us the wrong Fig 6. It is the same as Fig 5, but I don't think there were any alterations there in the revision.

RESPONSE: We apologize for this oversight and have now included the correct Fig 6 in this submission. The reviewer is correct, there were no changes in that figure.

--

Reviewer #3 (Remarks to the Author):

The authors have addressed all my concerns and the paper should now be accepted for publication.

RESPONSE: Thank you very much.

Reviewers' Comments:

Reviewer #1:

Remarks to the Author:

The authors have addressed most of my concerns. I ask them to reconsider one data set before they finalize their manuscript, about the data in Fig. 4IJK. Perhaps we do not understand each other. I called it "one replicate". The authors responded "each of the experiments... were performed three independent times."

1) If this replicate data exists, then show it. To be specific, the authors state that the red line for ey>hid,p35 is the same data in all three graphs (IJK). Where are the replicates? Show them either as error bars or as independent graphs in a supplemental figure.

2) The authors wish to separate the data into three panels to make their points, and find it clearer than the reviewer figure they provided where the data is all combined. I agree about clarity. But they must state in the legend that the same ey>hid,p35 data is shown in each panel, otherwise it is extremely misleading.

Quick question - in line 62, is it possible they mean IAP agonist (or activator) rather than antagonist?

Reviewer #2:

Remarks to the Author:

I still do not believe that this paper provides a sufficient novel advance to warrant publication in Nature Communications for the reasons outlined in my initial review. However, as the editor seems to want the paper, I hereby offer my input.

The manuscript has been improved but still has problems that require new data to be added as well as textual changes.

Figure 3, a-d, I apologize for missing the Laminin staining in the supplementary Figures. However, in these Laminin images in Figs S7a-d, Fig S12a-c as well as the Perlecan images in Fig 3a-d and 5a-c, what is shown is a picture only of the whole disk. The images that made Figure 1 so much clearer, are the b'-d' panels of the Perlecan staining which are 63x magnifications of the whole disk. These obviously match the quantitation in e. The SEM provides the final clear proof. In Fig 3a-d the Perlecan stain could perhaps match the quantitation in panel e although it isn't very clear, but in Fig 5 a-c the levels of Perlecan seem higher in a and b than in c from the pictures provided not matching the quantitation in d at all. Again in Fig S7a-d and Fig 12a-c of the Laminin, it seems as if there is a higher level of Laminin staining in the control, not lower as is shown in the quantitation in e and d, respectively.

The authors should at least provide a convincing close up picture for these Perlecan and Laminin stainings as they did for Fig 1, if they refuse to do the SEM requested by Reviewer 1 as a definitive measure.

The description of the quantitation in the Methods now more complete. This is very helpful for the readers of the paper.

Figure 3k Line 938 in their Figure legends the authors still say the HepCA + H2O2 treatment causes BM damage as indicated by the yellow arrowheads. Their quantitation argues otherwise, and they have stated that the damage is not significant in the main text in the rebuttal letter. To thus not confuse the readers, they should remove the yellow arrowheads and change the text.

Textual request 4)

To say two things correlate means that they have a mutual relationship or connection, that one depends upon the other. To say they coincide means they happen at the same time, with no connection implied. They have not demonstrated that BM damage and ROS correlate in Figure 2, for example by showing one depends upon the other or even showing spatial concordance. They never stained for Perlecan and DHE at the same time to allow such analysis to be conducted. They just show in Figure 2 that both BM damage and ROS happen in neoplastic tumor models and don't in hyperplastic models. They show in Figure 3b,c,i that though ROS is necessary, it is not sufficient to cause BM damage. I thus strenuously object to this Figure's implication in panel i that ROS production causes BM damage. They should change this panel to have ROS production and BM damage (or the lack thereof) each be the products of the respective tumor model (two arrows coming out from the one model) not one dependent on the other (one arrow after the other). Since they have also not in this Figure provided evidence that would allow them to say the two correlate, they should not use the word correlate in either the title of the Figure (line 905), in the title of this section in line 149, or in the description on page 183-7 which they have rewritten to include coincide as well as correlate, rather to remove "correlation" or "correlate" entirely. The reference to a correlative link between the two in line 198 is speculative rather than a conclusion, which I think is fine.

I think including Figure R1 in the paper as Fig S10 makes the story more comprehensible.

Their model in Figure 7 should indicate that there is another input along with ROS and JNK that is required for MMP2 induction and Basement Membrane damage given what they have shown in Fig 3 and Fig S10

Minor comment: Fig 1 h-j', yellow nearly impossible to see, use different color

POINT-BY-POINT RESPONSE

NCOMMS-19-25564B

Diwanji and Bergmann

Referee #1

The authors have addressed most of my concerns. I ask them to reconsider one data set before they finalize their manuscript, about the data in Fig. 4IJK. Perhaps we do not understand each other. I called it "one replicate". The authors responded "each of the experiments... were performed three independent times."

1) If this replicate data exists, then show it. To be specific, the authors state that the red line for *ey>hid,p35* is the same data in all three graphs (IJK). Where are the replicates? Show them either as error bars or as independent graphs in a supplemental figure.

Response: We initially showed the results of only one set of experiments because the results of all three experiments were very similar. We have now averaged them out with error bars. As shown in the new Figure 4, panels 4i,j,k, the results are very similar to the data we showed in the last submission:

Figure 4i

Figure 4j

Figure 4k

2) The authors wish to separate the data into three panels to make their points, and find it clearer than the reviewer figure they provided where the data is all combined. I agree about clarity. But they must state in the legend that the same *ey>hid,p35* data is shown in each panel, otherwise it is extremely misleading.

Response: We have added a sentence to the figure legend of Figure 4i-k that the *ey>hid,p35* data is the same in each panel.

Quick question - in line 62, is it possible they mean IAP agonist (or activator) rather than antagonist?

Response: No, we meant IAP-antagonist as Hid inhibits the IAP protein DIAP-1.

--

Referee #2

I still do not believe that this paper provides a sufficient novel advance to warrant publication in Nature Communications for the reasons outlined in my initial review. However, as the editor seems to want the paper, I hereby offer my input.

The manuscript has been improved but still has problems that require new data to be added as well as textual changes.

Figure 3, a-d, I apologize for missing the Laminin staining in the supplementary Figures. However, in these Laminin images in Figs S7a-d, Fig S12a-c as well as the Perlecan images in Fig 3a-d and 5a-c, what is shown is a picture only of the whole disk. The images that made Figure 1 so much clearer, are the b'-d' panels of the Perlecan staining which are 63x magnifications of the whole disk. These obviously match the quantitation in e. The SEM provides the final clear proof. In Fig 3a-d the Perlecan stain could perhaps match the quantitation in panel e although it isn't very clear, but in Fig 5 a-c the levels of Perlecan seem higher in a and b than in c from the pictures provided not matching the quantitation in d at all. Again in Fig S7a-d and Fig 12a-c of the Laminin, it seems as if there is a higher level of Laminin staining in the control, not lower as is shown in the quantitation in e and d, respectively.

The authors should at least provide a convincing close up picture for these Perlecan and Laminin stainings as they did for Fig 1, if they refuse to do the SEM requested by Reviewer 1 as a definitive measure.

Response: The higher Perlecan/Laminin appearance in Fig. 5a and 5b is because of the highly fluorogenic punctate-like staining at some areas, however, there is loss of staining in other areas that leads to the overall decrease in fluorescence intensity for 5a and 5b. Important is the overall homogenous appearance of Perlecan labeling in 5c which indicates an undamaged basement membrane. To make that clearer, we have now added close up (63x) images of Figs 3a-d, 5a-c, S7a-d and S12a-c. These new figures are shown below (panels 3a'-d', 5a'-c', S7a'-d' and S12a'-c' are the new 63x images) and in the manuscript:

New Figure 3 (panels 3a'-d' are the new 63x images; in the manuscript we have removed the prime labels)

New Figure 5 (panels a'-c' are the new 63x images; in the manuscript we have removed the prime labels)

New Figure S7 (panels a'-d' are the new 63x images; in the manuscript we have removed the prime labels)

New Figure S12 (panels a'-c' are the new 63x images; in the manuscript we have removed the prime labels)

The description of the quantitation in the Methods now more complete. This is very helpful for the readers of the paper.

Response: Thank you.

Figure 3k Line 938 in their Figure legends the authors still say the HepCA + H₂O₂ treatment causes BM damage as indicated by the yellow arrowheads. Their quantitation argues otherwise, and they have stated that the damage is not significant in the main text in the rebuttal letter. To thus not confuse the readers, they should remove the yellow arrowheads and change the text.

Response: We have removed the arrowheads and changed the text accordingly in line 938.

Textual request 4)

To say two things correlate means that they have a mutual relationship or connection, that one depends upon the other. To say they coincide means they happen at the same time, with no connection implied. They have not demonstrated that BM damage and ROS correlate in Figure 2, for example by showing one depends upon the other or even showing spatial concordance. They never stained for Perlecan and DHE at the same time to allow such analysis to be conducted. They just show in Figure 2 that both BM damage and ROS happen in neoplastic tumor models and don't in hyperplastic models. They show in Figure 3b,c,i that though ROS is necessary, it is not sufficient to cause BM damage. I thus strenuously object to this Figure's implication in panel i that ROS production causes BM damage. They should change this panel to have ROS production and BM damage (or the lack thereof) each be the products of the respective tumor model (two arrows coming out from the one model) not one

dependent on the other (one arrow after the other). Since they have also not in this Figure provided evidence that would allow them to say the two correlate, they should not use the word correlate in either the title of the Figure (line 905), in the title of this section in line 149, or in the description on page 183-7 which they have rewritten to include coincide as well as correlate, rather to remove “correlation” or “correlate” entirely.

The reference to a correlative link between the two in line 198 is speculative rather than a conclusion, which I think is fine.

Response: We appreciate the reviewer’s clarification. It is now clear what the reviewer means. As requested, we exchanged the word “correlate” with the word “coincide” wherever it appears in that section. We have also removed panel 2i from the manuscript to avoid further confusion.

I think including Figure R1 in the paper as Fig S10 makes the story more comprehensible.

Response: Thank you.

Their model in Figure 7 should indicate that there is another input along with ROS and JNK that is required for MMP2 induction and Basement Membrane damage given what they have shown in Fig 3 and Fig S10

Response: We have modified Figure 7 with an additional input, marked by an arrow and a “?” in the middle and right panels of 7b:

Minor comment: Fig 1 h-j', yellow nearly impossible to see, use different color

Response: That is a good point. We have replaced the yellow color Fig 1h-j' with magenta (see below):